# NeRAF: 3D Scene Infused Neural Radiance and Acoustic Fields

**Amandine Brunetto, Sascha Hornauer, Fabien Moutarde**
Center for Robotics, Mines Paris - PSL University
Paris, France
{amandine.brunetto,sascha.hornauer,fabien.moutarde}@minesparis.psl.eu

## Abstract

Sound plays a major role in human perception. Along with vision, it provides essential information for understanding our surroundings. Despite advances in neural implicit representations, learning acoustics that align with visual scenes remains a challenge. We propose NeRAF, a method that jointly learns radiance and acoustic fields. NeRAF synthesizes both novel views and spatialized room impulse responses (RIR) at new positions by conditioning the acoustic field on 3D scene geometric and appearance priors from the radiance field. The generated RIR can be applied to auralize any audio signal. Each modality can be rendered independently and at spatially distinct positions, offering greater versatility. We demonstrate that NeRAF generates high-quality audio on SoundSpaces and RAF datasets, achieving significant performance improvements over prior methods while being more data-efficient. Additionally, NeRAF enhances novel view synthesis of complex scenes trained with sparse data through cross-modal learning. NeRAF is designed as a Nerfstudio module, providing convenient access to realistic audio-visual generation. Project page: https://amandinebtto.github.io/NeRAF

## 1 Introduction

Sound is fundamental to human perception. Imagine standing in the middle of a bustling city. It's not only the visual cues but the sounds of car horns, footsteps, and chatter that guide your perception and decisions. Beyond conveying environmental details, sound provides crucial insights into context and atmosphere, enriching our understanding of the world in ways that sight alone cannot. In gaming and AR/VR simulations, audio is key to immersion, enhancing the realism of virtual environments and making experiences feel authentic.

Advances in novel view synthesis have enabled the generation of high-quality, photorealistic images from any camera position using a set of captured images (Mildenhall et al., 2020; Martin-Brualla et al., 2021; Zhang et al., 2020; Tancik et al., 2023), unlocking exciting applications for simulated environments. Yet, these methods lack acoustic synthesis, which is essential for fully immersive experiences. Learning scene acoustics presents additional challenges (Luo et al., 2022). As sound travels from its source to the listener it is influenced by the geometry of the space and the materials of objects and surfaces. Room Impulse Responses (RIRs) capture these complex interactions for specific source-listener pairs, but collecting RIRs is a laborious process, requiring sounds to be played and recorded at multiple positions.

To address this, recent research has focused on estimating RIRs at novel positions from sparse data (Majumder et al., 2022; Singh et al., 2021; Somayazulu et al., 2024). Inspired by NeRF's success in image synthesis, emerging studies (Luo et al., 2022; Su et al., 2022; Liang et al., 2023a;b) apply neural fields to learn the acoustic properties of an environment, allowing for audio synthesis from new source and microphone positions. However, these approaches often overlook the critical influence of 3D scene geometry on acoustics (Luo et al., 2022; Su et al., 2022; Liang et al., 2023a), or they rely on additional annotations that are unavailable in real-world scenarios (Liang et al., 2023b).

In response, we introduce NeRAF, a method that generates both novel views and RIRs at new sensor positions by jointly learning radiance and acoustic fields (Figure 1). NeRAF queries the radiance field to construct a voxel representation of the scene that encodes radiance and density. This grid

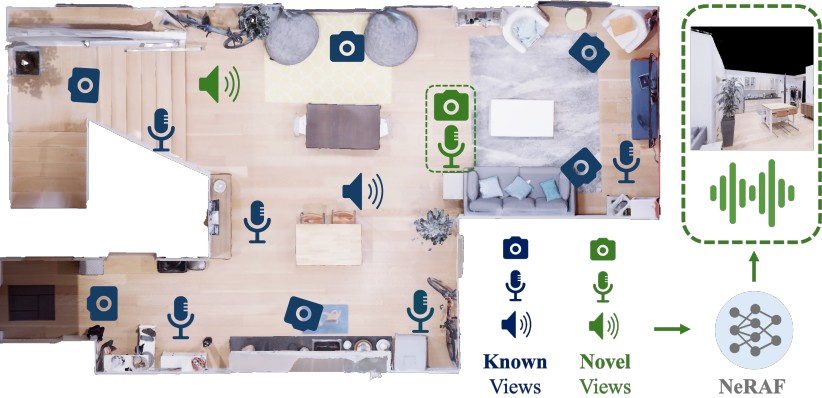

Figure 1: NeRAF synthesizes audio-visual data at novel sensor positions by learning radiance and acoustic fields from a collection of images and audio recordings. It enables audio auralization and spatialization, as well as improved image rendering, all of which are crucial for creating a realistic perception of space. NeRAF leverages cross-modal learning without the need for co-located audio and visual sensors for training. Our method allows for the independent rendering of each modality.

conditions the acoustic field with appearance and geometric 3D priors, without the need for additional annotations. NeRAF's cross-modal approach benefits both modalities: it achieves state-of-the-art RIR synthesis while being more data-efficient and enhances novel view generation in complex scenes. We validate our method on Sound Spaces (Chen et al., 2020a; 2022b), a simulated dataset, and on RAF (Chen et al., 2024), a real-world dataset. Nerfstudio (Tancik et al., 2023) is an easy-to-use, modular framework for NeRF. We design NeRAF as a new module to offer convenient access to audio-visual generation necessary for immersive experiences. Although the approach is cross-modal, each modality can be generated independently and the model can be trained without co-located audio and visual sensors, providing flexibility for various applications. We release NeRAF's code to the community.

## 2 RELATED WORKS

**Neural Radiance Fields.** NeRF (Mildenhall et al., 2020) synthesizes novel photorealistic views of a scene by modeling it as a continuous function that maps 3D spatial coordinates and viewing directions to radiance. Although it has demonstrated impressive results on objects and small bounded regions, it struggles with complex scenes where the camera can be oriented in any directions and content may exist at any distance. Building upon NeRF, (Martin-Brualla et al., 2021) handles unconstrained photo collections, enabling the use of diverse image sets. Other works improved NeRF to learn more complex scenes (Barron et al., 2022; Zhang et al., 2020) with higher efficiency and quality (Müller et al., 2022; Xu et al., 2022). Recently, Nerfstudio (Tancik et al., 2023) provided a modular framework that allows for a simplified end-to-end process of creating, training, and testing NeRF. It combines many existing NeRF improvements to create Nerfacto, a model suited for real static scenes. NeRAF is designed as a Nerfstudio module, bringing novel audio synthesis to existing state-of-the-art NeRF methods. Its cross-modal learning improves Nefacto performance on large complex scenes given a small image set.

**Neural Acoustic Fields.** Several works have extended the applicability of neural fields to the audio domain. Luo et al. (2022) (NAF) and Su et al. (2022) (INRAS) model acoustic propagation in a scene by learning an implicit representation that maps emitter-listener positions to RIRs. NAF conditions the acoustic field by learning local geometric information, while INRAS performs audio scene feature decomposition based on the acoustic radiance transfer model. NACF (Liang et al., 2023b) and AV-NeRF (Liang et al., 2023a) provide vision-based information to the acoustic field. NACF assumes the pre-acquisition of RGB-D images along with acoustic coefficients of surfaces. However, the latter is unavailable in real-life scenarios. AV-NeRF relies on a pre-trained Nerfacto model to render RGB-D information at the microphone's position. It then conditions the audio rendering at

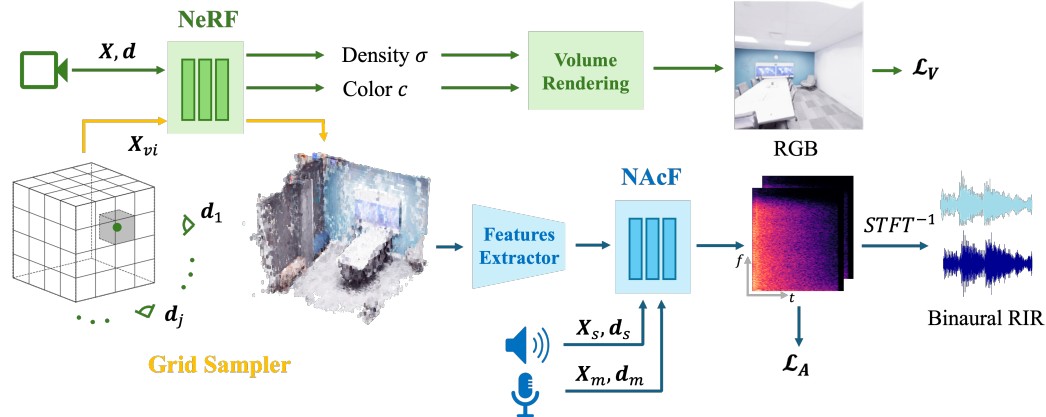

Figure 2: **NeRAF overview.** NeRF maps 3D coordinates, $\mathbf{X}$, and orientations, $\mathbf{d}$, to density and color. The grid sampler fills a 3D grid representing the scene by querying the radiance field with voxel center coordinates and multiple viewing directions. NAcF learns to map source-microphone poses and directions to STFT. It is conditioned by extracted scene features. Predicted RIRs can be convolved with audio to obtain auralized and spatialized audio matching the scene.

the same pose. In contrast, NeRAF extends beyond the camera's field of view by using NeRF to capture 3D scene information – without the need for more annotations – providing richer insights for the acoustic field. Contrary to AV-NeRF, our method does not need co-located audio and visual sensors for training. Therefore, it handles unequal amounts of training data for each modality, which is beneficial in real-world settings. Additionally, at inference, the vision modality is not necessary for audio rendering leading to greater versatility.

**RIR Synthesis.** RIRs have various applications, such as enabling immersive and spatialized audio for AR and VR, performing de-reverberation or providing insights about room acoustics. However, RIRs collection in real-world environment is time consuming and needs specialized hardware. Consequently, synthesizing them has been a longstanding research topic. Traditional methods rely on simulated approaches such as *wave-based* (Gumerov & Duraiswami, 2009; Thompson, 2006; Raghuvanshi et al., 2009; Bilbao et al., 2019) or *geometric* methods (Savioja & Svensson, 2015; Schissler & Manocha, 2016; Krokstad et al., 1968; Vorländer, 1989; Cao et al., 2016; Sprunck et al., 2022). While these methods effectively simulate sound propagation, *wave-based* methods face computation complexity and *geometric* methods struggle to simulate low-frequency acoustic phenomena such as interference and diffraction. Recent works have proposed to leverage the modeling ability of neural networks to learn room acoustics and estimate RIRs (Majumder et al., 2022; Luo et al., 2022; Su et al., 2022; Ratnarajah et al., 2021; 2022).

**Audio-Visual Learning.** Some works rely on both audio and visual information to perform acoustic related tasks such as audio binauralization (Gao & Grauman, 2019; Garg et al., 2023), auralization (Chen et al., 2022a; Somayazulu et al., 2024), de-reverberation (Chen et al., 2023a; Chowdhury et al., 2023) or RIR prediction (Singh et al., 2021; Majumder et al., 2022). Audio-visual learning has also demonstrated promising results for other tasks such as improving depth prediction (Christensen et al., 2020; Brunetto et al., 2023; Parida et al., 2021; Zhu et al., 2024), performing floorplan reconstruction (Majumder et al., 2023; Purushwalkam et al., 2021), navigating through a space (Younes et al., 2023; Gao et al., 2020; Chen et al., 2023c; 2020b; Gan et al., 2020) and estimating poses (Chen et al., 2023c).

## 3 METHOD

NeRAF generates both RGB images and binaural RIRs for any camera and microphone-source position and orientation. It achieves this by jointly learning neural radiance and acoustic fields. Although trained together, the audio and visual components can be used independently. The NeRAF pipeline, shown in Figure 2, consists of three modules: NeRF, the grid sampler, and NAcF. The NeRF learns the radiance field and generates RGB images. The grid sampler queries NeRF's radiance field

with each voxel's center, filling the 3D grid with color and density. NAcF then renders binaural RIRs, conditioned by the implicit appearance and geometric 3D priors derived from the features extracted from this 3D grid.

## 3.1 MODELING ACOUSTIC THROUGH RIRS

To model the acoustic of an environment, our method learns RIRs. They describe how sound propagates between one emitter and listener depending of the scene's geometry and material properties. For example, early reflections provide information about the distance to nearby surfaces, while late reverberation reflects the overall size and structure of the scene. Learning acoustics through RIRs offers several key benefits: (1) RIRs encapsulate the acoustic behavior between specific emitter and listener positions; (2) By convolving audio signals with RIRs, we can synthesize sound that mimics how it would be heard within the scene; (3) Binaural RIRs include head-related transfer functions (HRTFs), which are essential for accurately spatializing audio.

By learning to generate RIRs, our method captures essential sound propagation phenomena, enabling realistic audio synthesis with rich acoustic details.

## 3.2 NEURAL RADIANCE FIELD

Neural radiance field (NeRF) (Mildenhall et al., 2020) renders photo-realistic images from new view points. It represents a static scene as a 5D vector-valued continuous function whose input is a 3D location $\mathbf{X} = (x, y, z)$ and 2D viewing direction $\mathbf{d} = (\theta, \phi)$, and output is color $\mathbf{c} = (r, g, b)$ and volume density $\sigma$:

$$\text{NeRF} : (\mathbf{X}, \mathbf{d}) \rightarrow (\mathbf{c}, \sigma). \tag{1}$$

NeRF shoots rays $\mathbf{r}(t) = \mathbf{o} + t\mathbf{d}$ through image pixels from camera origin $\mathbf{o}$ between the near $t_{\text{near}}$ and far $t_{\text{far}}$ boundaries of the scene. 3D points $\mathbf{X}_n = \mathbf{o} + t_n\mathbf{d}$ are sampled along the ray and, along with $\mathbf{d}$ used as input of two MLPs. The first one maps $\mathbf{X}_n$ to the view-independent density $\sigma_n$ and a corresponding feature vector. The second MLP takes the feature vector and $\mathbf{d}$ to produce view-dependent color $\mathbf{c}_n$. The final color $C(\mathbf{r})$ of an image pixel are rendered using volume rendering equations:

$$C(\mathbf{r}) = \int_{t_{\text{near}}}^{t_{\text{far}}} \mathcal{T}(t_n)\sigma_n \mathbf{c}_n dt_n, \tag{2}$$

where $\mathcal{T}(t_n)$ is the transmittance expressed as $\exp(-\int_{t_{\text{near}}}^{t_n} \sigma_k dk)$.

NeRAF builds on Nerfacto (Tancik et al., 2023) as its NeRF backbone. Nerfacto integrates advancements from several works following (Mildenhall et al., 2020) including camera pose refinement (Wang et al., 2021; Lin et al., 2021), per image appearance conditioning (Martin-Brualla et al., 2021), proposal sampling (Müller et al., 2022), scene contraction and hash encoding. We selected Nerfacto for its robust performance in handling real-world static scenes. However, our method is NeRF-agnostic as it relies directly on the radiance field without altering the underlying NeRF model.

## 3.3 GRID SAMPLER

Sound propagation is omnidirectional and influenced by the 3D geometry of the environment. Therefore, we provide our neural acoustic field with 3D geometric and appearance priors about the scene.

To this end, we use our grid sampler, illustrated in Figure 3, to construct a 3D volume from the NeRF model. The scene is represented as a 3D grid of $S^3$ voxels with coordinates between 0 and 1 in accordance with NeRF's scene contraction. We shows in Section 4.3, that $S = 128$ leads to good performances and use it for all our experiments.

We populate the grid by sequentially selecting batches of voxel centers $\mathbf{X}_{vi}$ as queries to the radiance field. By processing voxels in order, we ensure that the grid is progressively updated to reflect the

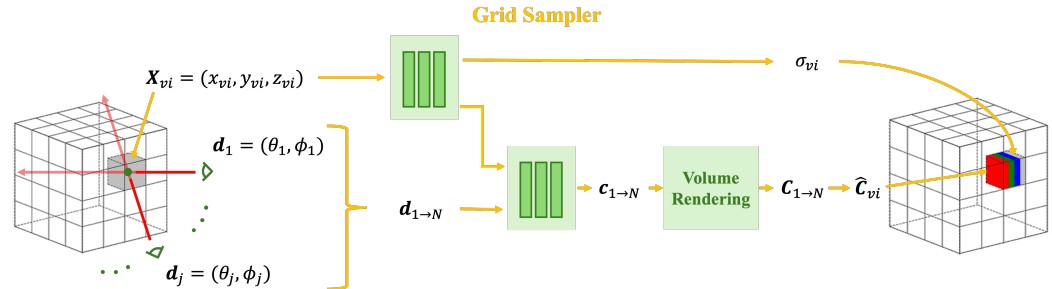

Figure 3: **Grid sampler.** We represent the scene as a grid of voxels. The grid is populated by querying the radiance field with the center coordinates of each voxel, $\mathbf{X}_{vi}$, along with multiple viewing directions, $\mathbf{d}_{1 \to N}$. We average the color values obtained from these directions. It results in a 7-channels 3D grid containing color $\widehat{\mathbf{C}}_{vi}$, density $\sigma_{vi}$ and the 3D coordinates of the voxel centers.

improvements in the NeRF model. To avoid slowing down training, we do not update the entire grid at each training iteration and process in batches. This sampling method guarantees complete grid population.

As NeRF color is view-dependent, we follow (Chen & Lee, 2023) and form $N = 18$ viewing rays per $\mathbf{X}_{vi}$. The goal is to obtain an average color of the voxel. The radiance field returns a density value $\sigma_{vi}$ and a color value $\mathbf{c}_{vi_j}$ per $j \in [1, N]$ viewing directions:

$$\text{NeRF} : (\mathbf{X}_{vi}, \mathbf{d}_{1 \to N}) \to (\mathbf{c}_{vi_{1 \to N}}, \sigma_{vi}). \tag{3}$$

For color, we apply the volume rendering equation to a ray containing a single point, $\mathbf{X}_{vi}$, with each viewing direction and average the values across these directions to obtain $\hat{\mathbf{C}}_{vi}$. For density, we compute the alpha compositing value $\alpha = 1 - \exp(-\sigma_{vi}\delta)$ where $\delta$ is a chosen small value. We populate the grid with these values and the 3D coordinates of the voxels, resulting in a 7-channel representation.

## 3.4 NEURAL ACOUSTIC FIELD

The goal of our neural acoustic field (NAcF), presented in Figure 4, is to learn a continuous neural representation of the scene's acoustics. To this end, NAcF maps microphone coordinates, $\mathbf{X}_m = (x_m, y_m, z_m)$, and orientations, $\mathbf{d}_m = (\theta_m, \phi_m)$, along with source position, $\mathbf{X}_s = (x_s, y_s, z_s)$, and orientation, $\mathbf{d}_s = (\theta_s, \phi_s)$, to the short-time Fourier transform (STFT) of a room impulse response:

$$\text{NAcF} : (\mathbf{X}_m, \mathbf{d}_m, \mathbf{X}_s, \mathbf{d}_s, t) \to \mathbf{RIR}(f, t). \tag{4}$$

The exact field parametrization adapted to the sensors' characteristics of each dataset is presented in Appendix A.

We learn RIRs through their STFT representation, resulting in STFT $\in \mathbb{R}^{C \times F \times T}$, where C, F and T denote the number of channels, frequency bins, and time bins, respectively. This is motivated by the fact that the time-frequency representation is smoother and more suitable to neural network prediction than the time domain (Luo et al., 2022). Thus, NAcF learns to infer log-magnitude STFTs. To better capture perceptual effects such as reverberation, we query NAcF with STFT time bins $t$. Consequently, NAcF outputs the $F$ frequency bins corresponding to the time query. We normalize $t$ between $[0, 1]$ and encode it with positional encoding. Similar to NeRF (Mildenhall et al., 2020) pose, microphone and source coordinates as well as direction are also embedded to a higher dimensional space using respectively positional encoding and spherical harmonic encoding.

Given the voxel grid, we concurrently train a ResNet3D (He et al., 2016) to extract relevant features for sound propagation. They serve as implicit indicators of scene's 3D geometry, which is central for acoustic modeling. Moreover, as shown in previous works (Chen et al., 2023b; Singh et al., 2021), information about materials can be inferred by a ResNet architecture from the scene's appearance and contribute to audio synthesis. We perform multimodal fusion by concatenating the features vector from the grid to the encoded poses, directions and the time query. Similar to NeRF, NAcF comprises two MLP blocks. The first block maps the input to a feature vector that embeds the input information and the acoustic scene. The second focuses on learning microphone specifics. For

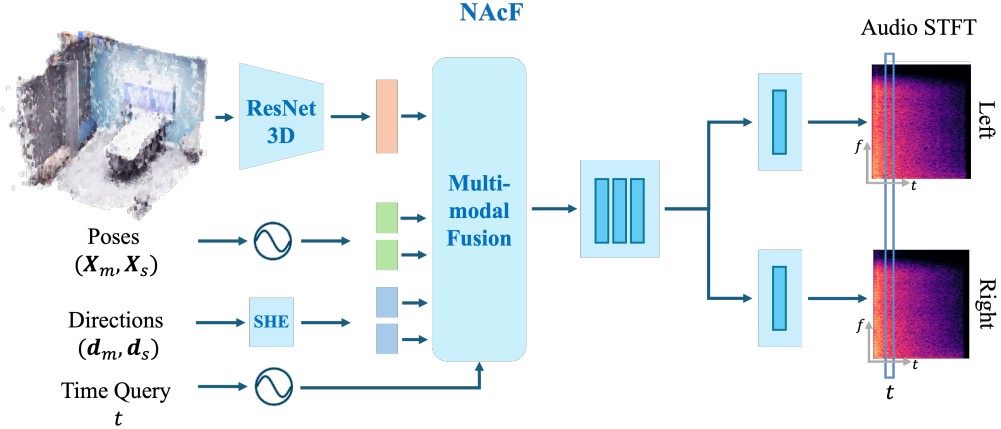

Figure 4: **Neural acoustic field.** NAcF maps microphone-source poses and directions to either binaural or monaural RIRs, with the number of output heads adjusted accordingly. NAcF is conditioned on scene features extracted through a 3D ResNet. Both poses and time queries are transformed into a higher-dimensional space using positional encoding, while directions use spherical harmonic encoding. The output of NAcF is a vector containing $F$ frequency values for each time query, $t$.

binaural microphones, this block learns the HRTF necessary for audio spatialization. It contains one MLP per microphone channel.

To render the complete STFT, we query all time bins $t \in [0, T]$. Finally, we use Griffin-Lim (Griffin & Lim, 1984; Perraudin et al., 2013) algorithm to obtain RIR waveforms from magnitude STFTs.

### 3.5 LEARNING OBJECTIVE

Cross-modal learning has proven beneficial in multiple works (Section 2). Thus, NeRAF trains jointly NeRF and NAcF. NAcF training is slightly delayed as it begins when the grid has been updated a few times. The feature extractor is jointly trained with the radiance and acoustic fields.

We do not modify the NeRF loss function $\mathcal{L}_V$. It usually consists in an MSE loss between ground-truth pixel color, $\mathbf{C}(\mathbf{r})$, and the rendered one, $\hat{\mathbf{C}}(\mathbf{r})$, and complementary losses, $\mathcal{L}_{\text{comp}}$, specific to each NeRF.

$$\mathcal{L}_V = \| \hat{\mathbf{C}}(\mathbf{r}) - \mathbf{C}(\mathbf{r}) \|_2^2 + \mathcal{L}_{\text{comp}}, \tag{5}$$

Inspired by (Yamamoto et al., 2019; 2020), we use a combination of spectral loss $\mathcal{L}_{\text{SL}}$ (Défossez et al., 2018) and spectral convergence loss $\mathcal{L}_{\text{SC}}$ (Arık et al., 2018) as our audio loss $\mathcal{L}_A$:

$$\mathcal{L}_A = \lambda_{\text{SC}} \mathcal{L}_{\text{SC}} + \lambda_{\text{SL}} \mathcal{L}_{\text{SL}}, \tag{6}$$

$$\mathcal{L}_{\text{SL}} = \|\log(|s| + \epsilon) - \log(|\hat{s}| + \epsilon)\|_2^2, \tag{7}$$

$$\mathcal{L}_{\text{SC}} = \frac{\| |s| - |\hat{s}| \|_F}{\| |s| \|_F}, \tag{8}$$

where $\hat{s}$ is the predicted STFT, $s$ the ground-truth STFT, $\|.\|_F$ denotes the Frobenius norm, $\|.\|_2$ the L2 norm and $\epsilon = 10^{-3}$. The values of our loss hyperparameters are found empirically and given in Appendix C. The spectral convergence loss emphasizes on spectral peaks helping especially in early phases of training while the spectral loss focus on small amplitude components which tends to be more important towards the later phases of training. Our loss differs from (Yamamoto et al., 2019; 2020) as we found out that the MSE spectral loss leads to a better trade-off between metrics than the L1 spectral loss (Su et al., 2022) and the MSE loss on magnitude STFT (Luo et al., 2022; Liang et al., 2023a). NeRAF results with those three losses are presented in Section 4.3. The complete learning objective can be resumed as:

$$\mathcal{L} = \mathcal{L}_V + \lambda_A \mathcal{L}_A. \tag{9}$$

Table 1: **Comparison with state-of-the-art.** Performance on the SoundSpaces dataset using T60, C50, and EDT metrics (Left) and on RAF (Right) with the same metrics and STFT error. Lower score indicates higher RIR quality.

| Methods | T60 (%) ↓ | C50 (dB) ↓ | EDT (sec) ↓ |
|---|---|---|---|
| Opus-nearest | 10.10 | 3.58 | 0.115 |
| Opus-linear | 8.64 | 3.13 | 0.097 |
| AAC-nearest | 9.35 | 1.67 | 0.059 |
| AAC-linear | 7.88 | 1.68 | 0.057 |
| NAF | 3.18 | 1.06 | 0.031 |
| INRAS | 3.14 | 0.60 | 0.019 |
| AV-NeRF | 2.47 | 0.57 | 0.016 |
| NACF | 2.36 | 0.50 | 0.014 |
| NACF w/ T | 2.17 | 0.49 | 0.014 |
| **NeRAF** | **2.14** | **0.38** | **0.010** |

| Methods | T60 (%) ↓ | C50 (dB) ↓ | EDT (sec) ↓ | STFT error (dB) ↓ |
|---|---|---|---|---|
| NAF | 10.08 | 0.71 | 0.021 | 0.64 |
| INRAS | 8.01 | 0.79 | 0.025 | 0.36 |
| AV-NeRF | 8.11 | 0.73 | 0.021 | 0.39 |
| **NeRAF** | **7.47** | **0.61** | **0.020** | **0.17** |
| NAF++ | 8.19 | 0.53 | 0.017 | 0.64 |
| INRAS++ | **6.17** | 0.57 | 0.017 | 0.39 |
| NACF | 6.62 | 0.59 | 0.017 | 0.39 |
| NACF w/ T | 7.31 | 0.59 | 0.018 | 0.39 |
| **NeRAF++** | 6.87 | **0.51** | **0.015** | **0.17** |

## 4 EXPERIMENTS

First, we benchmark NeRAF against related works (Luo et al., 2022; Su et al., 2022; Liang et al., 2023a;b) on the SoundSpaces and RAF datasets. Next, we demonstrate that cross-modal learning of the acoustic and radiance fields is also beneficial to vision, resulting in enhanced results for large complex scenes trained with sparse observations. Furthermore, we evaluate NeRAF's ability to generalize across multiple scenes. To assess the effectiveness of our method when trained with a limited number of audio recordings, we conduct a few-shot experiment. Finally, we investigate the impact of the 3D grid and our loss combination on NeRAF's performance. Video demonstrations showcasing NeRAF's audio-visual generation in both real and synthetic environments are available on our project page.

### 4.1 DATASETS

**SoundSpaces.** SoundSpaces (Chen et al., 2020a) is an audio simulator built upon Habitat Sim (Savva et al., 2019; Szot et al., 2021; Puig et al., 2024), a 3D simulator for embodied AI research. It provides binaural RIRs at discrete positions of a 2D grid with a spatial resolution of 0.5 m with four head orientations accompanied by sound source position. This popular benchmark (Gao et al., 2020; Chen et al., 2020b; Luo et al., 2022; Liang et al., 2023a; Su et al., 2022; Purushwalkam et al., 2021; Parida et al., 2021; Chen et al., 2022a) enables us to evaluate our method on diverse scene types while being under same settings as previous works. Following (Luo et al., 2022; Liang et al., 2023a; Su et al., 2022; Liang et al., 2023b), we select six indoor scenes from Replica (Straub et al., 2019). Two are small rooms (office 4 and room 2) with rectangular walls, two are medium room (frl apartment 2 and 4) with more complex layout and objects and two are complete apartments containing multiple rooms (apartment 1 and 2).

**RAF.** Real Acoustic Fields (RAF) (Chen et al., 2024) is a real-world dataset that includes high-quality, densely captured room impulse response data paired with multi-view images and precise 6DoF pose tracking data for sound emitters and listeners in two rooms. The room impulse response are recorded using the "earful tower", which is equipped with omnidirectional microphones. This dataset was recently used to benchmark previous works on real-world scenes.

**RWAVS.** RWAVS (Liang et al., 2023a) is a real-world dataset used to benchmark AV-NeRF. It does not contain RIRs but audio recordings. Our results on the RWAVS dataset are presented in Appendix J.

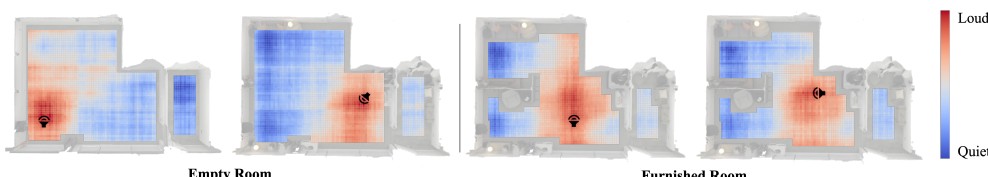

Figure 5: **Loudness maps visualization.** We visualize the intensity of predicted RIRs at different microphone positions for a given loudspeaker position and orientation. Intensities are averaged over multiple heights.

Table 2: **Quantitative results on cross-modal learning.** For highly complex scenes with very sparse training images, NeRAF audio-visual joint training improves vision performances, especially LPIPS. Evaluation on 50 test novel views. Delta are computed against Nerfacto (no audio).

| | PSNR ↑ | | | SSIM ↑ | | | LPIPS ↓ | | |
|---|---|---|---|---|---|---|---|---|---|
| # Images | 75 | 100 | 150 | 75 | 100 | 150 | 75 | 100 | 150 |
| *Apartment 1* | | | | | | | | | |
| Nerfacto | 22.48 | 24.77 | 27.00 | 0.838 | 0.882 | 0.902 | 0.246 | 0.161 | 0.118 |
| NeRAF | **23.33** | **25.07** | **27.70** | **0.858** | **0.884** | **0.915** | **0.216** | **0.140** | **0.105** |
| ΔNeRAF | +3.78% | +1.23% | +2.60% | +2.39% | +0.28% | +1.37% | -12.20% | -13.22% | -11.25% |
| *Apartment 2* | | | | | | | | | |
| Nerfacto | 19.10 | 22.67 | 25.62 | 0.777 | 0.836 | 0.884 | 0.381 | 0.232 | 0.160 |
| NeRAF | **20.53** | **23.46** | **26.06** | **0.802** | **0.856** | **0.895** | **0.322** | **0.206** | **0.150** |
| ΔNeRAF | +7.48% | +3.49% | +1.69% | +3.26% | +2.31% | +1.20% | -15.27% | -11.14% | -6.03% |

## 4.2 EVALUATION

**Metrics.** Following (Su et al., 2022; Liang et al., 2023a), we assess the quality of the predicted impulse responses using Reverberation Time (T60), Clarity (C50) and Early Decay Time (EDT). T60 reflects the overall sound decay within a room by measuring the time an impulse response takes to decay by $60\,\mathrm{dB}$. C50 measures the energy ratio between the first $50\,\mathrm{ms}$ of RIR and the remaining portion. It relates to speech intelligibility, and the clarity of acoustics. EDT focuses on early reflections. For RAF dataset, we also evaluate the STFT error which is the absolute error between the predicted and the ground-truth log-magnitude STFTs (Défossez et al., 2018; Luo et al., 2022; Chen et al., 2024). To assess the effect of cross-modal learning on novel view synthesis, we employ commonly used metrics: PSNR, SSIM, and LPIPS.

**Experimental Setup.** Similar to (Su et al., 2022; Luo et al., 2022; Liang et al., 2023a), we use $90\%$ of SoundSpaces audio data for training and $10\%$ for testing. They are pre-generated binaural RIRs with corresponding source and microphone positions. We resample the RIR to $22.05\,\mathrm{kHz}$ and compute STFT using $N_{\mathrm{FFT}} = W = 512$ and $H = 128$ where $N_{\mathrm{FFT}}$ is the number of FFT bins, W the Hann window size and H the hop length. To train the vision part, we generate RGB images using Habitat Sim. Camera poses and parameters were selected to limit the amount of visual data. For small, medium and large rooms, we use respectively 45, 75 and 150 observations. Evaluation set comprises 50 images. More details on the motivations and the generation process are available in Appendix D.

For RAF, we follow previous works experimental setup: we keep $80\%$ of data for training and $20\%$ for evaluation. Audios are cut to $0.32\,\mathrm{s}$, sampled to $48\,\mathrm{kHz}$ and processed into STFT using $N_{\mathrm{FFT}} = 1024$, $W = 512$ and $H = 256$. For the vision part, we randomly select one third of the data from cameras 20 to 25. Nerfstudio automatically keep $90\%$ of them for training and $10\%$ for evaluation.

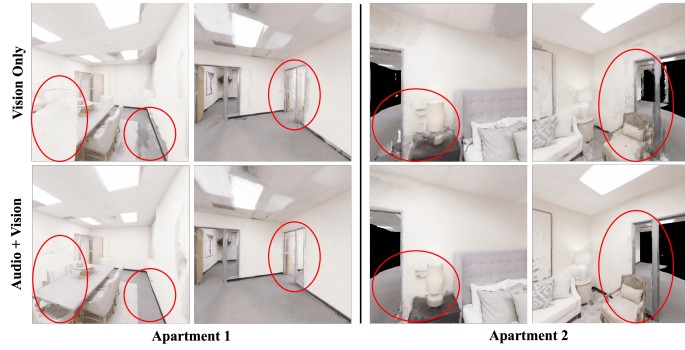

Figure 6: **Visualization of cross-modal learning impact on vision performances.** We compare Nerfacto (Vision Only) with NeRAF (Audio + Vision) on apartment 1 and 2. Learning both radiance and acoustic fields improves the quality of novel views synthesis, reducing floating artifacts and enabling more detailed images.

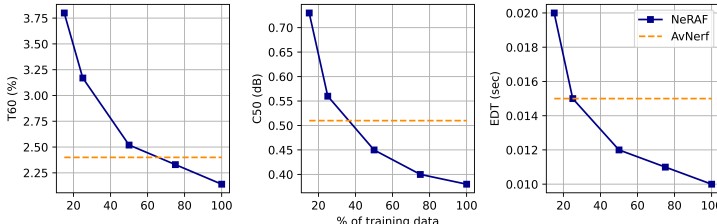

Figure 7: **Few-shot learning.** With 25% less audio recordings, NeRAF outperforms AV-NeRF trained on all the training data. Results are averaged on office 4, frl apartment 2 and apartment 2.

To ensure statistically significant results, we conduct each experiment 6 times and average results. More details on the implementation and architecture are presented in Appendix B and Appendix C.

**Results.** We compare NeRAF with state-of-the-arts methods on SoundSpaces in Table 1 (Left). Similar to previous works on neural acoustic field, we also compare our method to traditional audio encoding methods AAC and Opus (for Standardization, 2006; Foundation, 2012). AAC is a multichannel audio coding standard and Opus is an open audio codec. NeRAF significantly outruns existing methods for C50 and EDT. It achieves 22.4% improvement on C50 and 28.6% on EDT compared to NACF with temporal alignment. Compared to AV-NeRF – which is the method most similar to ours as it also employ Nerfacto and does not rely on a large amount of information known a priori – we achieve respectively 13.4%, 33.3% and 37.5% improvement for T60, C50 and EDT.

As SoundSpaces is a simulated dataset that presents limitations in term of visual and acoustics realism, we also present results on RAF in Table 1 (Right). NeRAF outperforms previous works best results, achieving -6.7% T60, -14.2% C50, -4.5% EDT and -52.5% STFT error. The second part of the table contains a comparison of methods using an additional energy decay loss term (Majumder et al., 2022; Liang et al., 2023b). While NACF initially relies on this term, other methods, with this new term, are denoted by "++". NeRAF++ still achieves state-of-the-art results, in particular for C50, EDT and STFT error. We present details about this loss in Appendix C. We showcase qualitative results of audio synthesis in Figure 5 and in Appendix I.

**Cross-Modal Learning.** We next investigate the impact of jointly learning acoustic and radiance fields on vision in complex scenes with very sparse training views, a challenging scenario where NeRF typically struggles. We train NeRAF and Nerfacto on the two large rooms. 75, 100, 150 images are used for training and the same 50 novel views are used for evaluation. In Table 2 we show that joint training of cross-modal learning improves novel view synthesis. It has a stronger impact on LPIPS which is related to the human-perceived similarity between two images. Overall, the performance improvement is more significant when the training images are very sparse. We present qualitative results in Figure 6: NeRAF reduces floating artifacts and enables more detailed images. Results on real scenes and additional visualizations are presented respectively in Appendix J and Appendix I.

**Generalization on Multiple Scenes.** We evaluate NeRAF's ability to generalize across multiple scenes. To this end, we train the acoustic field across multiple scenes without increasing the model size. Since Nerfacto is scene-dependent, we first train one NeRF-model for each scene. We populate a 128-grid per scene using the grid sampler and trained NeRFs. These grids are fed into the ResNet3D, which is trained

Table 3: **Multi-scene training.** Comparison of NeRAF performance when trained per scene versus on multiple scenes. Results are averaged across office 4, frl apartment 4 and apartment 2.

| Training | T60 (%) ↓ | C50 (dB) ↓ | EDT (sec) ↓ |
|---|---|---|---|
| Single-scene | 2.29 | **0.37** | **0.010** |
| Multi-scene | **2.24** | **0.37** | **0.010** |

jointly with the acoustic field. Batches are drawn from all scenes during training to ensure global convergence. Table 3 compares single-scene to multi-scene training. The results show similar performance, demonstrating NeRAF's capability to effectively learn the acoustic field across multiple scenes with a single model. In this setup, the visual modality supports the adaptation to different rooms. However, as NeRF and NAcF are not trained jointly, the visual component does not benefit from cross-modal learning. Detailed results for each scene are provided in Appendix O.

Table 4: **Ablation studies. Left:** Influence of the grid evaluated on RAF dataset. NG stands for no grid, 128 w/ 0 for a 128-grid filled with zeros, while 128 and 256 respectively correspond to grid of resolutions $S = 128$ and $S = 256$. Bold and underlined respectively indicates best and second best. **Right:** NeRAF's loss study. Performances averaged across SoundSpaces dataset.

| Grid | T60 (%) ↓ | C50 (dB) ↓ | EDT (sec) ↓ |
|------|-----------|------------|-------------|
| NG | 8.15 | 0.674 | 0.212 |
| 128 w/ 0 | 7.75 | 0.626 | 0.0205 |
| 128 | 7.47 | 0.609 | **0.0201** |
| 256 | **7.42** | **0.608** | 0.0204 |

| Loss | T60 (%) ↓ | C50 (dB) ↓ | EDT (sec) ↓ |
|------|-----------|------------|-------------|
| MSE | 1.89 | 0.53 | 0.015 |
| SC+SL$_{L1}$ | **1.56** | 0.42 | 0.013 |
| SC+SL$_{MSE}$ | 2.14 | **0.38** | **0.010** |

**Few-shot RIR Synthesis.** Previous works evaluated on SoundSpaces rely on a significant amount of RIRs, which is often unavailable in real-life scenarios. Therefore, we assess the performance of our model when trained with fewer data. We conduct a few-shot experiment on one small, one medium, and one large room and average the results. As shown in Figure 7, NeRAF is better than AV-NeRF on all metrics when trained with 75% of the data. It still outperforms AV-NeRF on EDT when trained with only 25% of the data. On RAF dataset – which is a real-world dataset that already contains a more realistic amount of training data – NeRAF still outperforms the previous methods when trained with 25% less data. Results on RAF are presented in Appendix E.

## 4.3 ABLATION STUDY

**Grid Impact.** We assess the grid impact on performances in Table 4 (Left) by first removing the grid (NG) and then filling the 128-grid with zeros (128 w/ 0). In both cases, NAcF is no longer conditioned by the 3D priors from the scene. Removing the grid also means we reduce the number of trainable parameters as we no longer keep the features extractor. By setting the grid to zero we remove scene-related information and still keep the exact same architecture. This way, we show that the performance gap cannot be explained only by an increased number of parameters but rather to the information extracted from the 3D-grid. We also study the impact of the resolution of the grid by comparing $S = 128$ and $S = 256$. Increasing the resolution to 256 improves T60 and slightly C50 but is accompanied by a minor EDT decline and increased training computation time.

**Loss Impact.** We examine the impact of our custom loss described in Section 3.5 compared to the MSE loss used in previous works (Liang et al., 2023a; Luo et al., 2022) and the combination with L1 spectral loss proposed by (Yamamoto et al., 2020) in Table 4 (Right). Our loss combination significantly improves C50 and EDT but decreases T60. Thus, this loss choice is a trade-off that can be adjusted to the final application. Note that NeRAF also achieves state-of-the-art performances with SC+SL$_{L1}$ combination.

## 5 DISCUSSION

**Limitation and Future Work.** While the neural acoustic field can be trained on multiple scene, it is still necessary to train NeRF separately for each scene. Future work should explore advancements in generalizable NeRF to develop a unified, generalizable approach for synthesizing audio-visual scenes. The current approach is also limited to static scenes. Addressing the challenge of learning implicit representations for audio-visual scenes with dynamic sound sources would be a significant advancement.

**Conclusion.** We introduce NeRAF, a cross-modal method that learns both neural radiance and acoustic fields. By conditioning the acoustic field with scene features from a 3D grid containing geometric and appearance information, NeRAF enables realistic audio auralization, spatialization and novel view synthesis. Our results demonstrate improvements over other methods, enhanced novel view synthesis in complex scenes and increased data efficiency.

ACKNOWLEDGMENTS

This work was supported by the French Agence Nationale de la Recherche (ANR), under grant ANR22-CE94-0003 and was granted access to the HPC resources of IDRIS under the allocation 2024-AD011015475 made by GENCI. We would like to thank Simon de Moreau, Raoul de Charette and the anonymous reviewers for their insightful comments and suggestions.

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

## A    FIELD PARAMETRIZATION

**SoundSpaces.**    SoundSpaces dataset lacks the complete acoustic field parameterization described in Equation (4). Microphone rotates only around the up-axis, $\theta$, and the source is omnidirectional. Thus, NAcF can be expressed as $(\mathbf{X}_m, \mathbf{d}_m = \theta, \mathbf{X}_s, t) \rightarrow (\text{RIR}_{\text{ch1}}, \text{RIR}_{\text{ch2}})(f, t)$.

**RAF.**    RAF dataset contains only monaural RIRs. Consequently, we replace the two heads presented in Figure 4 by a single head. The microphone is omnidirectional but the source is directional. We can therefore express NAcF as $(\mathbf{X}_m, \mathbf{X}_s, \mathbf{d}_s = \theta, t) \rightarrow \text{RIR}_{\text{mono}}(f, t)$.

## B    ARCHITECTURES

**NeRF.**    NeRAF works with any NeRF method without modifications. It only requires a radiance field that the grid sampler can query using voxel center coordinates and viewing directions. Our method only relies on NeRF to fill a voxel-grid representation of the scene. In this paper, we choose Nerfacto from Nerfstudio (Tancik et al., 2023). For more details on its architecture we invite readers to refer to Nerfstudio website and paper.

**Grid Sampler.**    The grid sampler creates a 3D grid of voxels with a given resolution. Grid coordinates are comprised between $[0, 1]^3$ fitting the scene contraction performed by NeRF. Consequently, knowing the exact dimensions of the room is not necessary. Voxels center coordinates and 18 viewing directions are queried to NeRF with no architecture modification. The grid is filled with color and density information along with voxels coordinates.

**NAcF.**    We train a ResNet3D-50 to embed the grid into features that best condition the acoustic field. For smaller scenes (office 4, room 2, apartment 2 and RAF dataset), we rely on 1,024 features. For larger scenes (frl apartment 2, 4, apartment 2) we use 2,048 features. At inference, only these features, and not the complete grid, are required to perform RIR synthesis. Note that we employ average pooling at the end of the ResNet, adjusting according to the grid size to consistently yield the desired number of features.

NAcF consists of 2 MLP blocks. The first MLP block takes as input the multimodal fusion of grid features, encoded positions, and directions obtained through vector concatenation. It comprises 5 layers, each followed by a Leaky ReLU activation with a slope of 0.1, and outputs a 512 intermediate representation of the acoustic field. The second MLP block takes this intermediate representation as input and predicts the $F$ frequencies corresponding to the time query, aiming to learn the microphone specifics (e.g., HRTF for binaural microphones in SoundSpaces). It consists of one separate MLP head per microphone channel. The final activation layer is a tanh function scaled between $[-10, 10]$ to fit the STFT log-magnitude range.

## C    IMPLEMENTATION DETAILS

**Hyperparameters.**    We implement our method using PyTorch framework (Paszke et al., 2019). We optimize NAcF using Adam optimizer (Kingma & Ba, 2014) with $\beta_1 = 0.9$ and $\beta_2 = 0.999$ and $\epsilon = 10^{-15}$. The initial learning rate is $10^{-4}$. It decreases exponentially to reach $10^{-8}$. For NeRF, just as AV-NeRF we keep default Nerfacto parameters.

For the first 2k iterations, we only train the NeRF part. It allows the grid to be filled and updated several times using batches of 4,096 voxel-centers. After, both NeRF and NAcF are train jointly. We use batch sizes of 4,096 for NeRF and 2,048 for NAcF. Note that NeRAF is trained by shuffling all STFT time slices in the train set. In NeRAF++, we require the full STFT to apply the energy decay loss so we train NeRAF by querying all the time bins necessaries to form the STFT. In that case we use a batch size of 34 STFTs. NeRAF is trained for 500k iterations but most runs reach their peak performance before, depending of the room size. We train our method on a single RTX 4090 GPU.

**Learning Objective Details.**    For the learning objective defined in Section 3.5, we empirically select $\lambda_A = 10^{-3}$, $\lambda_{SC} = 10^{-1}$ and $\lambda_{SL} = 1$.

To train NeRAF++, we add an additional energy decay loss term that reinforces the energy attenuation tendency of predicted RIRs. Note that unlike INRAS++ and NACF, we do not use multi-resolution STFT and only apply the loss to the predicted STFT. The audio learning objective becomes:

$$\mathcal{L}_A = \lambda_{SC}\mathcal{L}_{SC} + \lambda_{SL}\mathcal{L}_{SL} + \lambda_{ED}\mathcal{L}_{ED}. \tag{10}$$

with $\lambda_{SC} = 3$, $\lambda_{SL} = 1.5$ and $\lambda_{ED} = 5$.

We follow Liang et al. (2023b) formula. Given a magnitude STFT, $\mathbf{M}$, we first compute its energy in each time window by calculating the magnitude's square and aggregating it along the frequency dimension:

$$\mathbf{M}'^{(d)} = \sum_{f=1}^{F} \left( \mathbf{M}^{(f,d)} \right)^2, \ 1 \le d \le D \ , \tag{11}$$

where $\mathbf{M}' \in \mathbb{R}_{+}^{D \times 2}$ represents the energy in each time window. We then sum $\mathbf{M}'$ along the time dimension, aggregating the energy from the current step $d$ until the end $D$ for each time step $d \in [1, D]$ to capture the overall energy decay trend:

$$\mathbf{M}''^{(d)} = \sum_{i=d}^{D} \mathbf{M}'^{(i)}, \ 1 \le d \le D \ , \tag{12}$$

where $\mathbf{M}'' \in \mathbb{R}_{+}^{D \times 2}$. Finally, we measure the L1 distance between the ground-truth energy trend and the predicted one in the log space:

$$\mathcal{L}_{\text{ED}} = || \log_{10} \mathbf{M}''_g - \log_{10} \mathbf{M}''_p ||_1 \ . \tag{13}$$

**SoundSpaces.** To align with previous works on SoundSpaces, we resample all RIRs from 44,100 Hz to 22,050 Hz. We compute STFT with 512 FFT bins, a Hann window of size 512 and hop length of 128. We obtain log-magnitude STFT using $\log(|\text{STFT}| + 10^{-3})$. Like Luo et al. (2022), we cut STFT to a maximum length depending of the scene. If the STFT is shorter than the maximum length we pad it with its minimal value.

**RAF.** According to RAF (Chen et al., 2024), we cut audio to $0.32$ s and keep RIRs at 48 kHz. We compute STFT using 1,024 FFT bins, a Hann window of size 512 and hop length of 256.

**Queries Encoding.** We give the 3D position of microphone and source to NAcF. They are normalized using minimum and maximum values of the poses in the training set. We add a 1 m margin. Then, they undergo multi-scale positional encoding with $N = 10$ frequencies and maximum frequency exponent of $e = 8$. We obtain direction vectors from the microphone rotation and apply spherical harmonic encoding with 4 levels. The time queries are normalized within the range $[0, 1]$ and go through positional encoding with $N = 10$ and $e = 8$.

**Evaluation Metrics.** We inverse the transform of the predicted STFT using PyTorch Griffin-Lim algorithm running on GPU. We compute metrics against ground-truth waveforms.

To evaluate the quality of synthesized RIR waveforms $w$, we use T60, C50 and EDT errors obtained as follows:

$$\text{T60}(w, w_{\text{GT}}) = \frac{|\text{T60}(w) - \text{T60}(w_{\text{GT}})|}{\text{T60}(w_{\text{GT}})}, \tag{14}$$

$$\text{C50}(w, w_{\text{GT}}) = |\text{C50}(w) - \text{C50}(w_{\text{GT}})|, \tag{15}$$

$$\text{EDT}(w, w_{\text{GT}}) = |\text{EDT}(w) - \text{EDT}(w_{\text{GT}})|. \tag{16}$$

For RAF dataset, we follow previous works and also assess the quality of the time-frequency representation using the STFT error. To ensure fair comparison with methods directly predicting RIRs in the time domain, we perform the metric on the STFT obtained from the RIR in time domain. It means that we first go from the STFT predicted by NeRAF to a RIR waveform through Griffin-Lim algorithm and then compute its STFT before applying the metric. The STFT error is expressed as the absolute error between the predicted and the ground-truth log-magnitude STFTs:

$$\text{STFT error} \ (\mathbf{M}, \mathbf{M}_{GT}) = |\log(\mathbf{M}) - \log(\mathbf{M}_{GT})| \tag{17}$$

To evaluate NeRF performances we choose the commonly used metrics such as PSNR, SSIM and LPIPS.

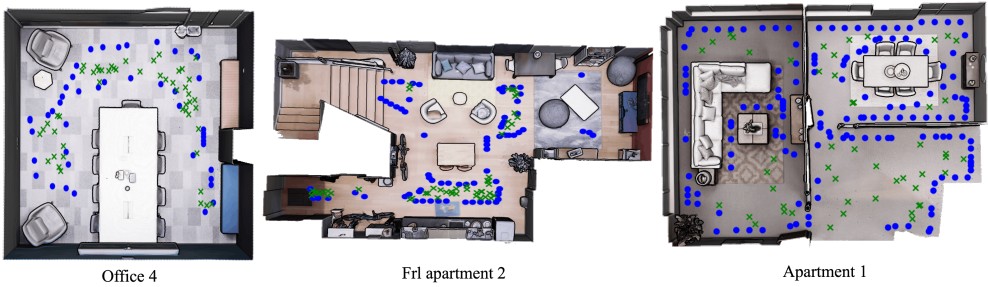

Office 4          Frl apartment 2          Apartment 1

Figure A1: Locations of visual observations for 3 SoundSpaces scenes. Blue dots corresponds to training views and green crosses to evaluation views. The exact poses coordinates are given in the supplementary material.

## D  SCENE VIEWS

**SoundSpaces.** SoundSpaces 1.0 provides a code to generate RGB and depth observations at microphone positions and orientations. They corresponds to $128 \times 128$ resolution images with a $60°$ field-of-view (FOV) sampled every 0.5 m with orientations $\theta \in [0°, 90°, 180°, 270°]$. In NeRAF, we decided to not rely on them for NeRF training and instead generate our own observations using Habitat Sim. This is motivated by real scenarios constraints. Views of the space may be obtained via a video recorded in the room. It is unlikely that someone will stop every 0.5 m and turn around to obtain images at each orientations. Moreover, those kind of views are not optimal for NeRF training leading, in the worst case, to the failure of less sophisticated methods, and, in the best case, to the use of more data. We also wanted to decorrelate microphone poses and camera poses for greater freedom in the usage of the method.

With Habitat Sim we rendered higher resolutions images with size $512 \times 512$ and with a $90°$ FOV. The larger FOV ensures more overlap between views. We randomly select positions at the edge of the room, orienting the camera to face its center with a random offset. For small room, medium and large room we use respectively 45, 75 and 150 training images. 50 test poses are randomly sampled in the room. Figure A1 presents examples of the positions obtained.

**RAF.** Real Acoustic Fields dataset captures high-quality and dense room impulse response data paired with multi-view images using VR-NeRF "Eyeful Tower" camera rig (Xu et al., 2023). It contains 22 cameras at different heights and pointing at different directions. We chose cameras 20 to 25 for training NeRAF's vision part. This choice ensures that we have sufficient overlapping views. The images have a 0.7 megapixels with resolution (i.e., $684 \times 1024$). For more details on those data we invite readers to refer to VR-NeRF paper. To train NeRAF, as we want to limit the number of images necessaries, we randomly sub-select one third of those images. We let Nerfstudio select 90% of them for training and 10% for evaluation.

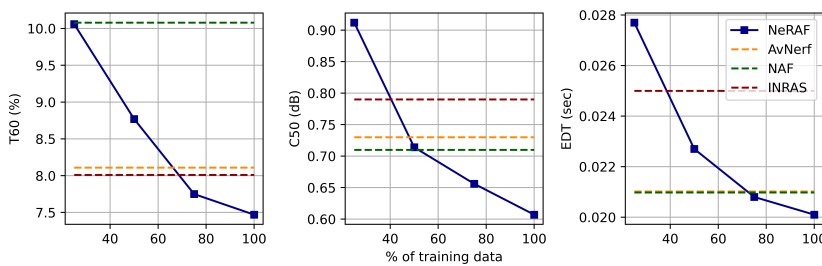

Figure A2: Evaluation of NeRAF on RAF with a decreasing number of training data.

Table A1: Audio model number of parameters along with storage and speed requirements at inference.

| Grid Features | Parameters (Million) | Storage (MB) | Speed (ms) |
|---|---|---|---|
| 1024 | 55.49 | 77.93 | 2.33 |
| 2048 | 71.89 | 97.84 | 2.64 |

## E  FEW-SHOT ON RAF.

RAF is a real-world dataset that contains a more realistic number of training data compared to SoundSpaces. We show in Figure A2 that NeRAF still outperforms INRAS, NAF and AV-NeRF with 25% less training data making it more data efficient.

## F  NUMBER OF PARAMETERS, STORAGE AND INFERENCE SPEED

We provide in Table A1 the numbers of parameters of the NeRAF audio model and evaluate its inference storage requirements and speed. We present results when the 1,024 and 2,048 features are extracted from the grid. For RAF dataset, all results presented in the paper are obtained using 1,024 features while for SoundSpaces half the scenes use 1,024 (office 4, room 2, apartment 2) and the other half 2,048 (frl 2, frl 4, apartment 1). The speed was averaged over the RAF dataset on a single RTX 4090. Note that at inference, the grid is not necessary and the extracted features are sufficient allowing the method to be faster and more compact. As our inference speed time is two orders of magnitude smaller than the length of the generated audio, NeRAF can be used for real-time applications.

## G  SOCIETAL IMPACT

By synthesizing realistic audio-visual scenes, NeRAF can enable immersive experiences in VR and gaming. Understanding acoustics is also useful for applications such as sound de-reverberation, source localization, and agent navigation. However, if misused, our method could contribute to the creation of misleading media, including the ability to mask and lie about someone's location.

## H  GRID VISUALIZATION

We display examples of sectional views of grids obtained using the grid sampler in Figure A3. They are then encoded into features using ResNet, which condition our neural acoustic fields.

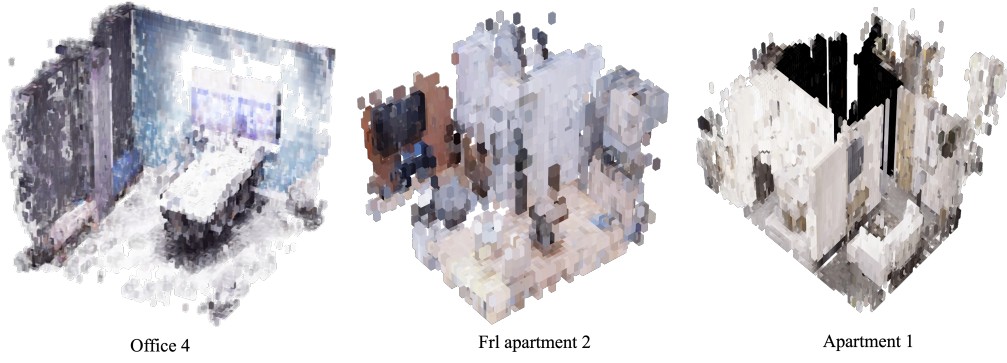

Office 4          Frl apartment 2          Apartment 1

Figure A3: Sectional views of $128^3$ grids obtained using the grid sampler.

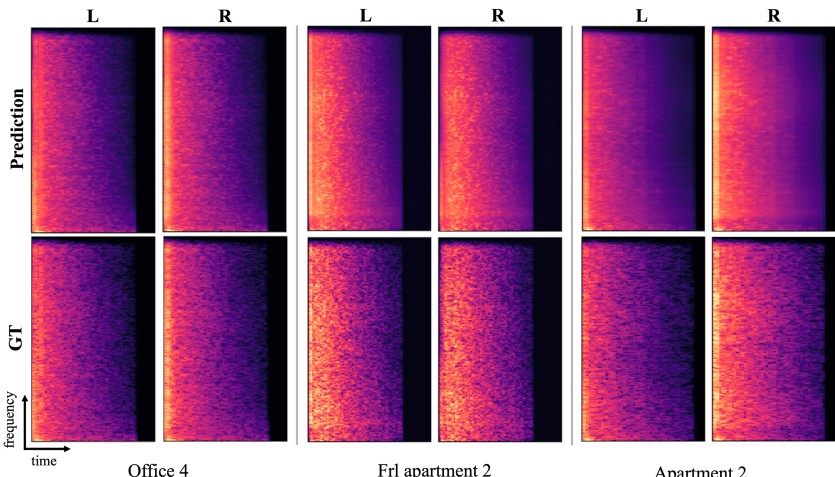

Figure A4: Examples of predicted log-magnitudes of binaural RIR STFTs compared to corresponding ground truth (GT). L and R respectively stand for left and right.

## I ADDITIONAL RESULTS VISUALIZATION

**Audio qualitative results.** We provide examples of log-magnitude STFTs predicted with NeRAF in Figure A4.

**Distance-aware Spatialized Audio.** We render binaural RIRs at various distances from the sound source and perform auralization by convolving an anechoic sound with these RIRs. This demonstrates one of the primary applications of NeRAF. In Figure A5, we illustrate that NeRAF is distance-aware: as the microphone approaches the sound source, the amplitude increases, and it decreases as the microphone moves away. Additionally, when the microphone is positioned to the right or left of the source, the audio channels' amplitudes reflect this spatial positioning.

**Video.** We provide video examples of NeRAF generation in supplementary material. The chosen audios are open source and sourced from AVAD-VR (Thery & Katz, 2019).

**Cross-modal Learning.** We provide additional qualitative results on the effect of joint learning neural acoustic and radiance fields in Figure A6.

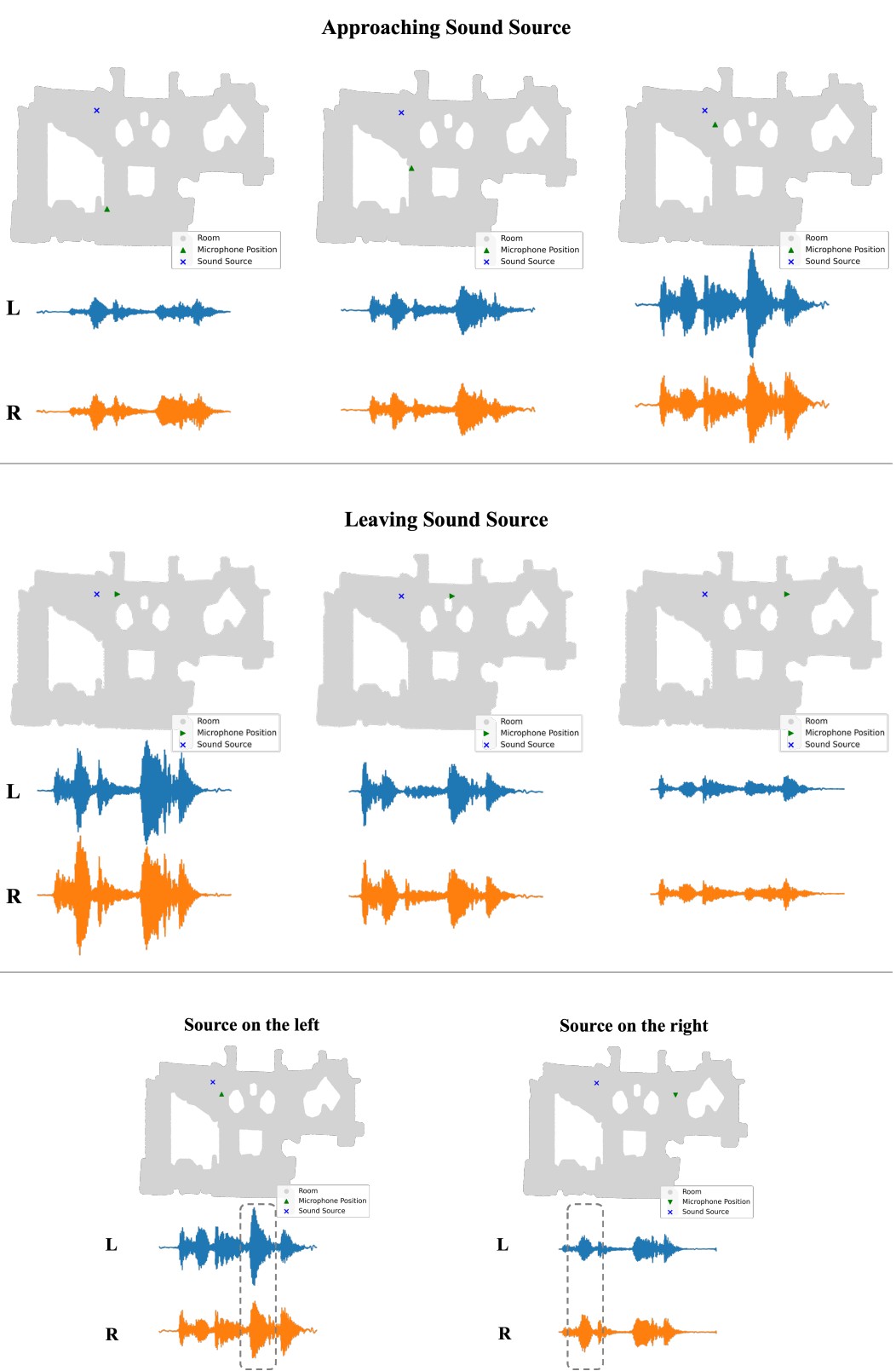

Figure A5: Examples of sounds generated with NeRAF on frl apartment 2. Our method successfully synthesizes spatialized and distance-aware sounds.

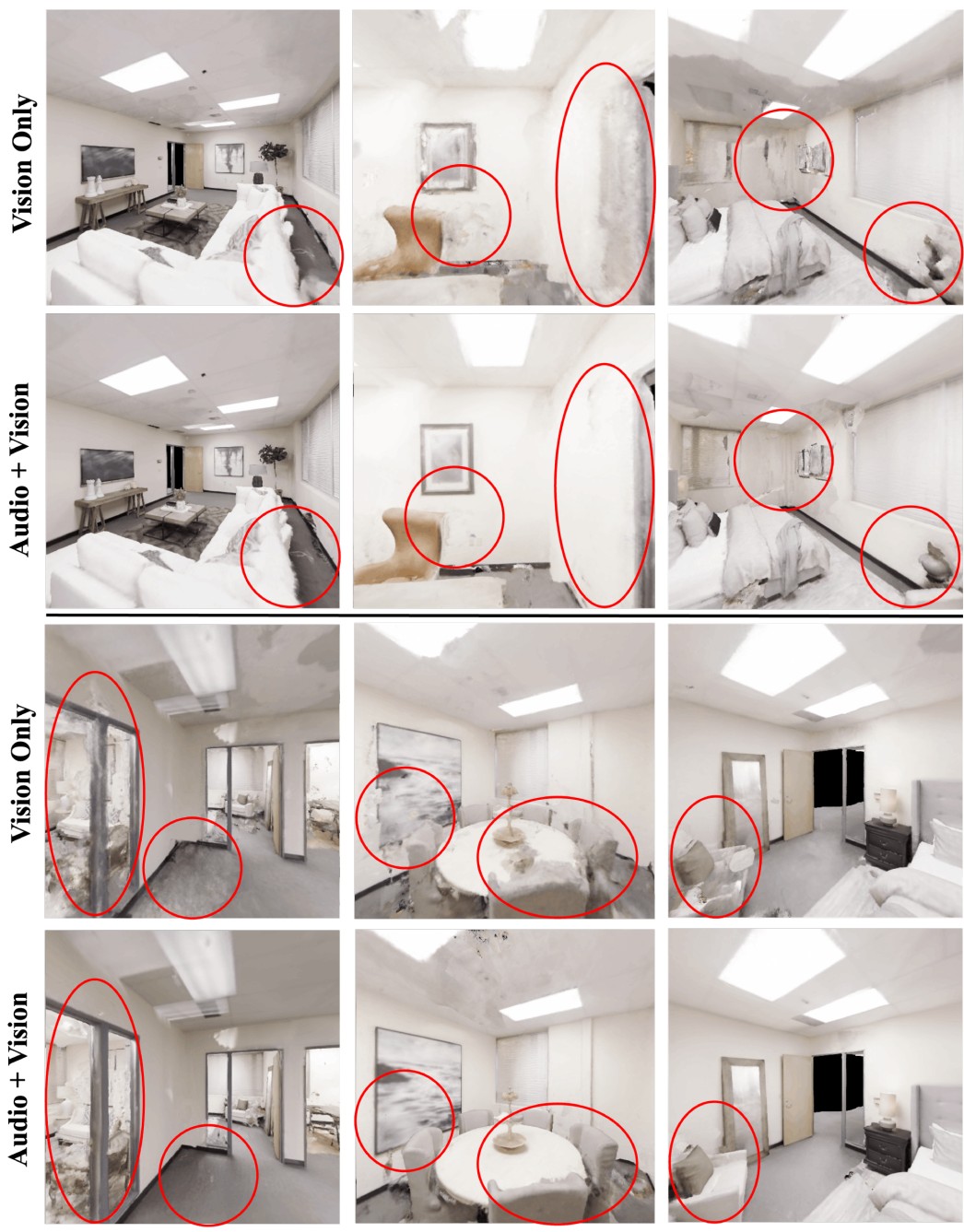

Figure A6: Additional visualization of cross-modal learning benefits on the visual modality.

Table A2: Comparison with state-of-the-art methods on RWAVS dataset. Bold is best, underlined is second best.

| Methods | Office | | House | | Apartment | | Outdoors | | Overall | |
|---------|--------|--------|-------|-------|-----------|--------|----------|--------|---------|--------|
| | MAG | ENV | MAG | ENV | MAG | ENV | MAG | ENV | MAG | ENV |
| INRAS | 1.405 | 0.141 | 3.511 | 0.182 | 3.421 | 0.201 | 1.502 | 0.130 | 2.460 | 0.164 |
| NAF | 1.244 | 0.137 | 3.259 | 0.178 | 3.345 | 0.193 | 1.284 | 0.121 | 2.283 | 0.157 |
| ViGAS [1] | 1.049 | 0.132 | 2.502 | 0.161 | 2.600 | 0.187 | 1.169 | 0.121 | 1.830 | 0.150 |
| AV-NeRF | 0.930 | 0.129 | **2.009** | **0.155** | **2.230** | **0.184** | 0.845 | 0.111 | **1.504** | 0.145 |
| NeRAF | **0.876** | **0.125** | 2.312 | 0.156 | 2.366 | **0.184** | **0.829** | **0.110** | 1.596 | **0.144** |

[1]Chen et al., Novel-view Acoustic Synthesis, CVPR 2023

## J RWAVS DATASET

Although we are aware AV-NeRF releases and evaluates its method on the RWAVS dataset, we chose to not present those results in the main paper. This is motivated by the difference between the tasks. NeRAF focus on learning RIRs, allowing the auralization of any given audio at inference. However, the RWAVS dataset does not contain RIRs but recordings of audio signals (e.g., music) played in a given environment. On this dataset, the network is therefore supposed to implicitly learn RIRs from them. To the best of our knowledge, it was never demonstrated that the representation learned can be used for the auralization of new audios.

Despite this, we present NeRAF performances on this dataset in Table A2. Similar to AV-NeRF, we adapt NeRAF to query frequencies instead of time and learn STFT masks. The same inverse modifications were done by AV-NeRF to evaluate on Sound Spaces (please see their supplementary material for more details). For every indoor scenes we extract 4,096 features from the grid and 1,024 for outdoors. This difference can be explained by the complexity of indoor scenes (e.g., many objects) compared to outdoors in this dataset. Because our loss combination is tailored for explicit RIR learning, we switch to MSE between magnitude STFTs.

Overall, NeRAF outperforms previous works tailored for RIR predictions and outperforms AV-NeRF in office, outdoors and slightly on apartment (ENV). We believe NeRAF is less performant on House because of the unusually large number of small objects in the scene. It would need a very high-resolution grid to capture them.

RWAVS contains very complex environments on which we can show that NeRAF cross-modal training benefits vision. Results are presented in Table A3. This demonstrates that learning both acoustics and radiance fields is interesting on both real and simulated data.

Table A3: Quantitative results on RWAVS large complex scenes with sparse training data.

| | PSNR ↑ | | SSIM ↑ | | LPIPS ↓ | |
|---------|--------|--------|--------|--------|--------|--------|
| Training Images | 1/6 | 1/4 | 1/6 | 1/4 | 1/6 | 1/4 |
| House | | | | | | |
| Nerfacto | 18.20 | 18.70 | 0.674 | 0.708 | 0.339 | 0.270 |
| NeRAF | **18.43** | **18.85** | **0.685** | **0.710** | **0.324** | **0.264** |
| ΔNeRAF | +1.26% | +0.81% | +1.65% | +0.25% | -4.31% | -2.14% |
| Apartment | | | | | | |
| Nerfacto | 19.14 | 19.64 | 0.746 | 0.776 | 0.251 | 0.211 |
| NeRAF | **19.24** | **19.68** | **0.748** | **0.778** | **0.249** | **0.210** |
| ΔNeRAF | +0.51% | +0.23% | +0.23% | +0.23% | -0.96% | -0.34% |

## K    RATIONALE FOR USING 3D PRIORS

Unlike light, sound propagates as a spherical wavefront in 3D from any point of emission. This means that sound waves spread outward omnidirectionally and interact with the entire 3D room's geometry, bouncing off surfaces, being absorbed, and reflecting throughout the space. Even with a directional source, which emits sound more strongly in certain directions, the propagation remains omnidirectional in space, influenced by environmental interactions rather than constrained by the initial emission direction.

Previous works (Liang et al., 2023a;b) rely on images (2D) to provide scene information to the acoustic field but we argue that conditioning on 3D information better suits the properties of sound propagation. For example, features like reverberation (linked to the room volume and surfaces) depend on complete 3D volume, not just what is visible in front of the microphone or camera. By leveraging 3D information, acoustic fields can capture the acoustics of the scene more effectively.

In this paper, we propose to use NeRF as a convenient way for providing 3D scene priors – geometry and appearance – directly to the acoustic field without needing explicit 3D annotations, such as meshes or point clouds, which are complex to obtain.

## L    IMPACT OF FURNITURES

We assess the impact of furnitures nearby a microphone on the predicted RIR. We conducted a first study on the apartment 1 and apartment 2 scenes from SoundSpaces, which both include furnished and spacious areas. Given a source, comparing performance across these areas did not reveal a clear trend; differences were observed in both directions. To extend the analysis, we averaged performance across all microphones within each area, regardless of the source. We also conducted this study on a real-scene, Furnished Room, from the RAF dataset. Again, results varied by scene, with no consistent pattern (see Table A4). Thus, we find no correlation between a microphone's proximity to furnitures and prediction accuracy.

Table A4: Performance comparison for microphones located near furniture (Furnished) and in spacious areas (Spacious). No correlation is observed between a microphone's proximity to furniture and prediction accuracy.

| Area | T60 $\downarrow$ | C50 $\downarrow$ | EDT $\downarrow$ |
| --- | --- | --- | --- |
| *Apartment 1* | | | |
| Furnished | **2.689** | 0.598 | 0.0165 |
| Spacious | 2.810 | **0.521** | **0.0116** |
| *Apartment 2* | | | |
| Furnished | **2.382** | **0.493** | 0.0138 |
| Spacious | 3.470 | 0.566 | **0.0102** |
| *Furnished Room* | | | |
| Furnished | 7.630 | **0.624** | 0.0171 |
| Spacious | **6.721** | 0.626 | **0.0156** |

## M    UNKNOWN SOURCES

To evaluate NeRAF's ability to generate RIRs at sources pose unseen during training, we create a new split of the RAF dataset with 85% of sources for training and 15% for testing, maintaining a similar data volume per split as in the original split. Indeed, the original RAF's split includes overlap in sources between training and testing, though not in source-microphone pairs. This aligns with previous works focusing on RIR prediction at new source-microphone pairs instead of unknown sources (see Table 1). On the unseen source split, NeRAF achieves T60 = 8.04, C50 = 0.76, EDT = 0.028 and STFT error = 0.171, suggesting that NeRAF can generate RIR for unseen sources.

## N    SEPARATE TRAINING OF EACH MODALITY

As shown in Table 2 and Figure A6, joint training improves image generation in large, complex scenes with sparse visual observations. Here, we asses the impact of separate training on the audio modality. In this scenario, a NeRF (without NAcF) is fully trained and used to populate a 3D grid. Subsequently, this grid is used to train NAcF without the NeRF component. Audio generation performance remains comparable for both approaches, which is expected since the acoustic field in both setups leverages a 3D grid representation of the scene. Regarding image generation performance, for smaller scenes with sufficient observations, such as office 4, there is no notable difference in visual performance between joint and separate training. Detailed results can be found in Table A5.

Table A5: Comparison of joint training vs. separate training for the audio and visual components. Joint training improves visual performance in complex scenes with limited training data, while audio generation remains comparable in both cases.

| Training | T60 ↓ | C50 ↓ | EDT ↓ | PSNR ↑ | SSIM ↑ | LPIPS ↓ |
|---|---|---|---|---|---|---|
| *Office 4* | | | | | | |
| Joint | **1.22** | **0.25** | **0.007** | **25.99** | **0.921** | 0.081 |
| Separate | **1.22** | 0.26 | **0.007** | 25.94 | 0.916 | **0.076** |
| *Apartment 1* | | | | | | |
| Joint | 2.73 | **0.57** | **0.015** | **27.70** | **0.915** | **0.105** |
| Separate | **2.71** | **0.57** | **0.015** | 27.00 | 0.902 | 0.118 |

## O    MULTI-SCENE TRAINING

To supplement the results presented in Table 3, we provide an evaluation of the multi-scene model on each individual scene. The model was trained across three scenes: office 4, frl 4, and apartment 2, and evaluated on each of these scenes. We compare these results with those obtained from single-scene training in Table A6. Each grid is encoded into 1,024 features using ResNet3D.

Table A6: Comparison of NeRAF performances when trained on multiple scenes against single scene.

| Training | T60 ↓ | C50 ↓ | EDT ↓ |
|---|---|---|---|
| *Office 4* | | | |
| Single-scene | **1.22** | **0.25** | **0.007** |
| Multi-scene | 1.37 | 0.28 | **0.007** |
| *Frl 4* | | | |
| Single-scene | 2.76 | **0.31** | **0.010** |
| Multi-scene | **2.74** | **0.31** | **0.010** |
| *Apartment 2* | | | |
| Single-scene | 2.88 | 0.54 | **0.013** |
| Multi-scene | **2.62** | **0.53** | **0.013** |
| *Mean* | | | |
| Single-scene | 2.29 | **0.37** | **0.010** |
| Multi-scene | **2.24** | **0.37** | **0.010** |

