# OpenReview forum: "NeRAF: 3D Scene Infused Neural Radiance and Acoustic Fields"
_ICLR.cc/2025/Conference — ICLR 2025 Poster_

### Official Review · Reviewer_DBxo · 2024-10-28

**Soundness:** 3
**Presentation:** 2
**Contribution:** 3
**Rating:** 6
**Confidence:** 5

**Summary:**

This paper introduces a novel method that simultaneously learns acoustic and radiance fields for a recently introduced problem, novel acoustic view synthesis. The core idea involves integrating 3D scene features when rendering room impulse responses for the acoustic field. Experiments conducted on both a synthetic dataset using SoundSpaces and the recently introduced real-world dataset, RAF, demonstrate the effectivenss of the proposed method over previous methods.

**Strengths:**

- The paper tackles an interesting and novel problem, and the concept of jointly learning acoustic and radiance fields is well-motivated, as visual and acoustic information can complement each other to learn spatial cues.

- Although AV-NeRF also utilizes vision to aid in learning an acoustic field, the proposed approach uniquely learns a radiance field and introduces a grid-sampler to extract visual features, which the acoustic field is conditioned upon.

- Table 1 demonstrates that the proposed method achieves notable improvements compared to AV-NeRF, INRAS, etc.

- It's good to see that the proposed method is evalauted on a real-world dataset, apart from simulated data in SoundSpaces. The authors also faithfully evaluate on the RWAVS dataset and reported the results, despite the different settings of the tasks.

**Weaknesses:**

- In supplementary video, it only shows the comparisons of the proposed method againt audio-only baseline, which is a weak baseline? It would be compare the qualitative differences between the proposed method and the closest performing prior work (AV-NeRF or NACF), to demonstrate the quantitaive improvement is perceptually meaningful.

- The main new component is the grid sampler component that extracts scene features to condition the RIR rendering. How beneficial is the joint training? What if separately training the visual component and a fixed grid sampler? How useful is the scene feature from grid sampler? What if a single image is provided as input and a ResNet 2D is used? How helpful is conditioning the task on 3D scene information?

- Following on the above questions, some additional ablations could be informative, such as

1. Comparing joint training vs. separate training of visual and acoustic components

2. Evaluating the impact of using a fixed vs. trainable grid sampler

3. Comparing 3D scene features from the grid sampler vs. 2D features from a single image

4. Quantifying the benefit of 3D scene conditioning vs. no scene conditioning

**Questions:**

Please see above.

---

> ### Author Response · Authors · 2024-11-18
> **Response to reviewer DBxo**
>
> We thanks reviewer DBxo for the encouraging remarks and feedback.
>
> ## Weaknesses
> **(W1)** In the supplementary video, all examples except the last showcase both sound and visuals generated by NeRAF. The last one, i.e. audio-only, illustrates NeRAF’s capacity to render sound and vision independently while being jointly-trained; it is not intended as a baseline comparison.
>
> We agree that comparing directly against prior methods would strengthen the qualitative assessment. However, AV-NeRF and NACF do not currently offer accessible code for RIR synthesis. AV-NeRF released code specifically for RWAVS, which lacks RIR data and NACF, which does not offer any code, is limited to audio synthesis. Both methods do provide sample videos of their own method on their respective project page, but the aforementioned constraints limit direct side-by-side comparisons in our video.
>
> **(W2, W3.4)** To assess how useful the 3D scene conditioning from the grid sampler is, we conducted an ablation study (Table 3). Results show that (1) removing the grid sampler (no grid, no ResNet) significantly reduces performance, and (2) replacing the NeRF-generated grid with an empty grid (filled with zeros) also lowers performance, even when keeping the ResNet3D architecture intact. This demonstrates that a meaningful 3D grid provides valuable scene information that enhances acoustic modeling, beyond merely increasing the model’s parameter count.
>
> **(W2, W3.1)** Joint training improves image generation in large, complex scenes with sparse visual observations, as shown in Table 2 and Figure 6. For smaller scenes with sufficient observations, such as office 4, there is no notable difference between vision performances for joint and separate training. Audio generation performance remains equivalent for both approaches, which is expected since the acoustic field in both setups leverages a 3D grid representation of the scene. Detailed results can be found in the table below and we added this experiment in appendix.
> | **Scene**       | **Training**        | **T60**            | **C50**            | **EDT**              | **PSNR**             | **SSIM**             | **LPIPS**            |
> |-----------------|---------------------|--------------------|--------------------|----------------------|----------------------|----------------------|----------------------|
> | **Office 4**    | Joint | **1.22**  | **0.25**  | **0.007** | **25.99**  | **0.921**  | 0.081 |
> |  | Separate | **1.22** | 0.26 | **0.007** | 25.94 | 0.916 | **0.076** |
> | **Apartment 1** | Joint | 2.73 | **0.57** | **0.015** | **27.70** | **0.915** | **0.105**     |
> | | Separate | **2.71** | **0.57** | **0.015** | 27.00 | 0.902 | 0.118   |
>
> **(W2, W3.3)** While AV-NeRF and NACF uses 2D images to provide scene information for the acoustic field, we argue that conditioning on 3D information better suits the properties of sound propagation. Sound propagates in all directions as a 3D wavefront, interacting with the entire room's geometry, bouncing off surfaces, being absorbed, and reflecting throughout the space. For example, features like reverberation (linked to the room volume and surfaces) depend on complete 3D spatial data, not just what's visible in front of the microphone. By leveraging 3D information, NeRAF captures the acoustics of the scene more effectively, as evidenced by its superior results over 2D-based approaches like AV-NeRF. We added a section about rationale for 3D priors in the appendix.
>
> **(W3.4)** The grid sampler has two main steps: (1) Grid population with fixed density and color values, representing geometry and appearance. The grid does not contain trainable parameters. (2) Grid encoding using a ResNet3D, which is the only trainable part, responsible for compressing these features into a smaller vector for the acoustic model. If the ResNet3D were not trainable, it would lack the ability to effectively encode and compress the scene information. We also envisioned to use a pre-trained ResNet 3D but to best of our knowledge there is none that is available for voxels.

---

### Official Review · Reviewer_h8tx · 2024-10-29

**Soundness:** 2
**Presentation:** 2
**Contribution:** 2
**Rating:** 6
**Confidence:** 4

**Summary:**

This paper introduces NeRAF, a new method for simultaneously learning acoustic fields and visual radiance fields. The authors demonstrate that this joint training approach enhances the quality of both generated room impulse responses and RGB images. Additionally, the proposed method offers flexibility, allowing for independent generation of single modalities during inference. Experiments conducted on both simulated (SoundSpace) and real-world (RAF) datasets show that NeRAF outperforms state-of-the-art methods in both audio and visual generation tasks.

**Strengths:**

- In general, I like the proposed multi-modal joint training approach that is appealing for its concise design and its potential to enhance the generated quality of both visual and acoustic fields. The flexibility to leverage single modalities when needed is also an advantage.
- The paper is generally straightforward to follow. Extensive studies have been conducted on both synthetic and real datasets, along with healthy supplementary materials, code, and a demo video.

**Weaknesses:**

- The task of rendering on both visual and acoustic signals has been previously explored (e.g., AV-Nerf), thus limiting the novelty of this work. From a technical perspective, the primary distinctions between NeRAF and AV-Nerf lie in 1) the use of 3D scene features for RIR generation and 2) the joint training pipeline. However, these technical contributions are not particularly significant to me.
- More analyses on cross-modal learning are needed. Table 2, Figure 6, appendix I and J show that joint training of cross-modal learning improves novel view synthesis but it is unclear whether there is any improvement in acoustic field synthesis.
- It is unclear why 3D ResNet is the best scene feature extractor. Some ablation studies would be helpful here.
- The loudness maps in Figure 5 exhibit some non-smooth textures. Why does it happen? Can NeRAF effectively represent a continuous acoustic field?
- NeRAF demonstrates better performance on RIR and image generation metrics, yet significant challenges persist. These include multi-scene representations, multiple or/and unknown sources, discrepancies between synthetic and real-world scenarios, and inverse problems. While this paper showcases improved training data efficiency, the impact is limited due to the density of synthetic datasets like SoundSpace. Even at 20%, the volume of training data remains substantial.
- From the perspective of human perception, it is unclear whether users can perceive actual difference between NeRAF and AV-NERF.
- NeRAF is obviously not a small model (Table A1). Especially it is still for single scene purpose.
- Some minor comments:
  - there was no context for Nerfstudio module until L90. Please clarify and add references in abstract and intro.
  - typo: wrong subscript $\sigma_{vi}$ in Eqn 4
  - In demo video, it would be better to provide trajectories in real scenes as well.

**Questions:**

- Why choosing magnitude spectrogram + griffin-lim instead of directly optimizing on waveform?
- Acoustic data in SoundSpace is actually in 2D (RIRs are the same for different heights). How does the 3D grid features help? Are 2D features good enough?
- In the demo video, are all videos shown on the right synthesized?

---

> ### Author Response · Authors · 2024-11-18
> **Response to reviewer h8tx (p1/2)**
>
> We thank reviewer h8tx for the thoughtful review.
>
> ## Weaknesses
> **(W1)** Although AV-NeRF introduced the use of vision to enhance neural acoustic learning, NeRAF advances the idea in several key ways:
> - **3D Spatial Awareness**: AV-NeRF relies on a pre-trained NeRF to generate 2D images for each microphone pose, providing only information about the scene directly in front of the microphone to the acoustic field and thus limiting spatial awareness. In contrast, NeRAF uses a 3D representation that provides full scene geometry information, for example room dimensions and 360° structure. This aligns well with the 3D nature of sound propagation and our results show improved accuracy of acoustic modeling.
> - **Joint Training Benefits**: Unlike AV-NeRF, which does not improve the visual modality, NeRAF’s joint training enhances NeRF performance in complex, sparsely observed scenes (Table 2, Figure 6, Appendix J) while also achieving SOTA RIR generation (Table 1).
> - **Efficient Inference**: AV-NeRF requires rendering an image for each RIR prediction, slowing inference and limiting real-time applications. NeRAF eliminates this bottleneck, as it decouples the two modalities at inference: the NeRF (and ResNet) is no longer needed for audio generation.
> - **Flexible Data**:  AV-NeRF requires co-located audio-visual data for training, where camera images match the exact positions and orientations of microphones. NeRAF, however, can be trained with separate audio and visual data, making it more versatile and practical for real-world applications - as underlined by Y2XC (S2). Additionally, NeRAF is more data efficient, achieving the same accuracy with 25% less data.
>
> **(W2)** Cross-modal learning benefits acoustic synthesis by providing 3D scene information to the acoustic field through vision, eliminating the need for additional data like meshes, which are difficult to obtain. We show in Table 3 that a grid populated via the radiance field, enabled by cross-modal learning, improves RIR synthesis accuracy.
>
> **(W3)** We selected 3D ResNet as our scene feature extractor primarily due to its compatibility with voxel-based data, its widespread use, and its simplicity. We acknowledge that other architecture could be interesting and plan to explore more advanced and lightweight architectures in future works.
>
> **(W4)** In Figure 5, RIRs were generated for microphones spaced 0.1m apart on a 2D plane, with loudness values represented as discrete colored squares. Despite this visualization artifact, NeRAF can represent a continuous acoustic field, as it allows querying RIRs at any sensor pose: it learns implicitly the continuous function that maps microphone and source poses to RIRs (Equation 5). Still, some errors in the acoustic prediction may arise because, as any model, NeRAF can make mistakes. We have no ground truth to evaluate it quantitatively but results align with the room geometry.
>
> **(W5)** We agree that key challenges remain: sound propagation is a very complex phenomenon, and implicitly learning the acoustics of a scene is not a straightforward task.
> Multi-scene representation is an interesting future work that could also help leverage synthetic data for fine-tuning on limited real-world data, bridging the gap between synthetic and real scenarios.
> We do not currently have access to a dataset with multiple sources playing at the same time, so we cannot assess NeRAF in this case.
> We recognize that SoundSpaces is a high-density synthetic dataset, which is why data efficiency is a key focus of our study. Our experiments also demonstrate NeRAF's resilience to reduced training data in real-world settings (Figure A2).
> We are grateful for the reviewer’s insights and will address these limitations in future works.
>
> **(W6)** We thank the reviewer for raising the important point of human perception, which is the objective we must not lose sight of. Evaluating perceptual differences between NeRAF and AV-NeRF would ideally involve a dedicated human study to assess spatialization, realism, and audio-visual coherence. Unfortunately, conducting such a study is outside the scope of this work. However, our evaluation relies on established metrics that, while objective, correlate with aspects of human perception. These metrics suggest a perceptual benefit over AV-NeRF, although direct human validation would provide more insights.
>
> **(W7)** We acknowledge that NeRAF is not a small model. That said, it still achieves real-time inference (Table A1) and compress the acoustic scene more compactly than non-neural methods [1].
> While this work focuses on single-scene learning, we are actively investigating ways to scale NeRAF to multi-scene use cases. Our future works will also explore reducing model size while maintaining or improving performance to enhance scalability and efficiency.
>
> **(W8)** We thanks the reviewer for the comments, we took them into account in the updated paper version.

---

> ### Author Response · Authors · 2024-11-18
> **Response to reviewer h8tx (p2/2)**
>
> ## Questions
> **(Q1)** We chose magnitude spectrograms with the Griffin-Lim algorithm for the following reasons: (1) Optimizing directly on waveforms, especially at high sampling rates (e.g., 48kHz), is computationally expensive and time-consuming. Spectrograms are more compact, allowing for faster processing. (2) Using spectrograms enables the network to focus on high-level features, such as spectral and temporal patterns, rather than learning all the raw waveform details. (3) The time-frequency representation has more structure and is smoother making it more suitable to neural network prediction than the raw waveform [1]. (4) The Griffin-Lim algorithm efficiently reconstructs waveforms from spectrograms, offering a good balance between audio quality and computational cost.
>
> **(Q2)** While SoundSpace 1.0 data is captured with sensors at fixed heights, sound still propagates in 3D. The figure 2 in SoundSpaces paper [2] highlights how 3D room structures influence the sound propagation in the simulator. This is achieved by relying on bi-directional path tracing that models sound reflections in the room’s 3D geometry [2]. We also double-checked with Sound Spaces 2.0 [3] – which is relying on the same mechanism but allows for configurable simulation parameters such as sensor poses – that RIR recorded with different heights were different, further underlining the 3D propagation in the simulator. Therefore, having 3D scene information from the grid, such as the volume and geometry of the space, provides valuable context for the acoustic field. This 3D knowledge helps the model better understand the scene's acoustic characteristics, going beyond the limitations of 2D features. We added a section on rationale for 3D features in appendix.
>
> **(Q3)** In the demo video, all the content, including both sound and visuals, has been generated with NeRAF. The only exception is the final video, where only the sound is synthesized. Its goal is to illustrate that each modality can be rendered independently, if wanted.
>
> **References:**
> [1] “Learning neural acoustic fields”, Luo et al., NeurIPS 2022
>
> [2] “SoundSpaces: Audio-Visual Navigation in 3D Environments”, Chen et al., ECCV 2020
>
> [3] “SoundSpaces 2.0: A Simulation Platform for Visual-Acoustic Learning”, Chen et al., ArXiv 2022

---

### Official Review · Reviewer_v8xr · 2024-11-02

**Soundness:** 2
**Presentation:** 3
**Contribution:** 2
**Rating:** 5
**Confidence:** 5

**Summary:**

The paper introduces a method that jointly learns acoustic and radiance fields named as NeRAF, which can synthesize both novel views and spatialized room impulse responses at arbitrary target viewpoints. They claim by conditioning the acoustic field on 3D radiance field, NeRAF can benefit both visual and acoustic modalities. Experiments on both synthetic and real datasets show NeRAF generates high-quality audio and improves performance on both novel view synthesis and acoustic modeling tasks.

**Strengths:**

+ The paper aims for a very interesting topic: joint learning of visual and acoustic rendering for 3D scenes. The authors proposed an effective pipeline to solve this challenging task, and explore how the two modalities can combine together.

+ Thorough experiments are conducted on both synthetic and real datasets, comparing with reliable baselines and plenty of ablations to show the effectiveness of the proposed modules. Especially for the grid sampling visualization, from which we can see what physical meaning the grids have learnt through the joint training.

+ The paper is well organized and easy to follow with clear presentation.

**Weaknesses:**

- Weak cross-modality learning. The proposed cross-modality learning way for acoustic learning is to use ResNet3D to extract 3D geometric and appearance priors, and then inject it for acoustic rendering. However, I doubt it due to two aspects:
    - The model is scene-dependent, which means fits one scene per model. In this case, it's hard to guide the ResNet3D to truly focus on properties that affect acoustic learning and physically correspond to geometry/materials etc., especially without any explicit supervision on it. The model can easily just take the grid sampling extractor as a large latent feature generator to help the model memorize the data distribution in the room.
    - From visualization of grids in Figure A3, we can see the grids feature is more correlated to visual rendering task, which makes sense since visual rendering has strong supervision on the grids directly (density and color etc.). However, this can be tricky if we want to understand how the two modalities are actually integrated. The grids seem to fail to correlate with acoustics. According to the first aspect, strong evidence is needed here to support the two modalities are actually helping each other in a way that has claimed physical meanings (instead of just more latent features that benefits memorizing the scene)

- One way that may help to really use the visual modality is to do multi-conditioning training, i.e. one model fits multiple scenes. In this case, the visual can be helpful to adapt to different rooms, and we may be able to observe how audio and visual modality integrate. With real visual representation, the model can do more than just fitting single room.

- If the visual and audio modality are helping each other in a pure latent way without any physical meaning, the soundness of claimed motivations and contribution will be largely weaken.

- The grids are in the fixed shape, which might be hard to fit large room, where the resolution can be super coarse and may degrade the performance.

**Questions:**

Please see the weaknesses above.

---

> ### Author Response · Authors · 2024-11-18
> **Response to reviewer v8xr**
>
> We thank reviewer v8xr for the thoughtful comments.
>
> ## Weaknesses
> **(W1 & 3)** The grid already contains 3D geometric and appearance priors as it has been populated with density (related to geometry) and color (related to appearance) by querying the radiance field with voxel center coordinates. The ResNet3D role is solely to encode the large 128^3 grid into a smaller feature vector for the acoustic field.
>
> The reviewer is right, as the ResNet is trained for each scene, it is not explicitly trained to extract physically meaningful data such as materials. Thus, we can only affirm that it is encoding the scene in a way that best helps acoustic field learning. We made sure that the grid sampling module is not acting as a large latent feature generator for the model to memorize the data distribution in the room:
>
> - The grid is non-trainable and therefore cannot memorize acoustic-specific data.
> - To ensure ResNet3D is not merely acting as additional parameters for memorization, we performed an ablation study (Table 3) where the grid was filled with zeros, removing all geometric and appearance information, but keeping the same ResNet3D architecture. Results showed reduced performance, confirming that ResNet3D effectively leverages the grid to improve acoustic learning. While we cannot specify the exact features extracted, this ablation provides evidence that ResNet3D focuses on acoustic-relevant information.
>
> The visual appearance of the grid in Figure A3 aligns with visual rendering since it only contains density and color data from NeRF. This setup is intentional and provides 3D information without requiring additional annotations such as meshes.
>
> We would like to complete with details on how the audio modality help the visual one. Since the audio loss backpropagates through the grid and thereby influences the radiance field, we examined its impact on NeRF performance. As shown in Figure 6 and Table 2, incorporating the audio modality improves NeRF results in complex scenes with sparse observations, indicating that the two modalities support each other meaningfully.
>
> **(W2)** We thank the reviewer for this valuable suggestion. We plan to tackle multi-conditioning training of the acoustic field in future works. Time is short but we will try to explore it during the rebuttal period.
>
> **(W4)** We agree that the voxelized grid has inherent resolution limits, especially in larger or cluttered rooms. Currently, the grid resolution represents a trade-off between precision and computational complexity. Improving this representation for better scalability is a priority for future works.

---

### Official Review · Reviewer_Y2XC · 2024-11-02

**Soundness:** 2
**Presentation:** 3
**Contribution:** 3
**Rating:** 5
**Confidence:** 4

**Summary:**

The paper presents NeRAF, a novel framework for synthesizing both vision and spatial audio in 3D scenes (spatial audio is the main focus of this paper). NeRAF’s key innovation is its ability to jointly learn visual neural radiance fields and spatial acoustic fields. The approach leverages cross-modal learning, meaning it incorporates visual features to enhance audio field prediction, and vice versa. It improves efficiency and realism in generating room impulse responses (RIRs). Unlike previous methods, NeRAF does not require aligned audio-visual data, making it more versatile and data-efficient. The method achieves state-of-the-art results in accuracy and quality of audio rendering, particularly in benchmarks like SoundSpaces and the Real Acoustic Fields (RAF) dataset.

**Strengths:**

1. **Enhanced Realism**. NeRAF uniquely integrates visual and acoustic information, using visual scene features to inform the acoustic model and vice versa. This cross-modal approach improves the fidelity of both visual and acoustic outputs, leading to more realistic audio-visual experiences without the need for additional aligned datasets.

2. **Data EVersatility**: Unlike other methods requiring dense, aligned audio-visual data, NeRAF operates effectively with sparse datasets and does not rely on co-located audio-visual sensors for training. This flexibility makes it suitable for various real-world applications, especially where dense data collection is impractical.

3. **State-of-the-Art Performance**: This paper demonstrates measurable improvements over existing methods in key metrics like Clarity (C50), Reverberation Time (T60), and Early Decay Time (EDT), setting a new standard for realistic audio-visual scene synthesis.

**Weaknesses:**

1. **Problem Setting Correctness**: In section 3.4, the authors write both the audio source and two microphones' (two ears) have directions. If so, it contradicts with the claim that: Sound propagation is omnidirectional (L205), because sound propagation from directional source isn't uniformally directional and thus may not be measured by RIR (the situation becomes more complex if the microphones are also directional). Moreover, the SoundSpaces 1.0 data provided RIR are emitted by point source without specific direction. The authors should 1. provide more detail discussion and theoretical proof showing how RIR they learned fits for directional source and microphones setting. 2. More detail regarding how directional information are used in SoundSpace 1.0 data or their own Habitat-sim simulated data (like how to set the direction and how to ensure the simulated RIR is conditioned on the set direction).

2. **Motivation Concern**: The task of this paper is to learn an acoustic field that can predict binaural RIR, but this paper proposes to learn auxiliary visual NeRF together with the acoustic field learning. One question naturally arises is that the authors should show why learning NeRF is essential for learning acoustic field, through either ablation study or theoretical analysis. If crossmodal visual learning is essential to improve acoustic field learning, is there any other more light weight visual learning mechanism other than NeRF can also improve the acoustic field learning. Given the title "3D scene infused neural radiance and acoustic fields" emphasizes the neural radiance field, more detailed discussion and investigation on the impact of visual radiance field on acoustic fields are required.

**Questions:**

1. See weakness section.

2. Experiment: since the framework is scene-specific where the train and test are based on the same room, how do the authors guarantee the train and test data are different? Another important discussion, if we fix a source position, what the predicted RIR accuracy difference on microphone position close to furniture and position on spacious area?

---

> ### Author Response · Authors · 2024-11-18
> **Response to reviewer Y2XC**
>
> We thank reviewer Y2XC for the valuable feedback.
>
> ## Weaknesses
> **(W1)** In section 3.4, we define the task in a general way: As RIRs are defined relative to a microphone and a source, we parametrize the NAcF such as it maps their pose and directions to a RIR. However, none of the dataset have both a directional microphone and a directional source. Consequently, we keep the direction only if the sensor has a directional directivity pattern. For each dataset, the adapted field parametrization equation is presented in Appendix A. We modified Section 3.4 to clarify this point.
>
> In line 205, we are referring to the physical property of sound propagation: it naturally propagates as a spherical wavefront in 3D from any point of emission. While a directional source emits sound more strongly in a certain direction, the propagation itself remains omnidirectional in space, influenced by environmental interactions rather than constrained by the initial emission direction.
>
> To address your specific points:
>
> (1) To learn the RIRs that fit sensors specifics, we compute the audio loss directly with the ground truth recorded using each dataset’s source and microphone characteristics. This ensures that the model accurately fits the directionality and settings of the original setup. For binaural microphones, the supplementary videos and figure A5 also illustrates NeRAF’s capability to generate spatialized RIRs.
>
> (2) We are using the SoundSpaces 1.0 pre-generated RIR. They are indeed recorded with an omnidirectional source and we do not add a direction. Thus, we discard the direction of the source in Equation 5 and NAcF parametrization becomes (Xm, dm, Xs, t) → RIR(f,t).
>
> **(W2)** Other visual mechanism than NeRF can be envisioned to improve acoustic field learning. 2D visual cues can improve acoustic field learning [1]. They are easy to acquire and process but they lack 3D scene geometry that plays a major role in acoustics. For example, room volume and surfaces strongly influence reverberation but are challenging to capture in 2D. While 3D annotations like meshes could provide this information, they are more complex to obtain. NeRF can be used to provide 3D scene prior (geometry and appearance) to the acoustic field without needing explicit 3D or extra annotations. This is done by populating a 3D grid with density and color information learned through NeRF. The benefits of this setup on the acoustic field are demonstrated in our grid ablation study (Table 3). It is also worth mentioning that while the first NeRF methods were long to train, more recent methods such as Nerfacto are very fast to converge and thus not a bottleneck for the method. The bottleneck lies in 3D processing that demands more heavy models than 2D. To limit this, NeRAF is designed for efficient inference: once trained, only the compact encoded grid vector is needed for generating audio, without requiring the NeRF and ResNet3D. We plan in future works to explore lighter 3D processing techniques.
>
> ## Questions
> **(Q1)** For RAF dataset, we use the train-test split provided by the authors on GitHub. For SoundSpaces, we follow previous works by using 90% of the pre-generated RIRs for training and 10% for evaluation, ensuring no overlap between these sets. We will release the exact RIR indices for each set to the community.
>
> **(Q2)** We thank the reviewer for the suggestion. We conducted the proposed study on the Apartment 1 and Apartment 2 scenes from SoundSpaces, which both include furnished and spacious areas. Comparing performance across these areas for a fixed source did not reveal a clear trend; differences were observed in both directions. To extend the analysis, we averaged performance across all microphones within each area, regardless of the source. We also conducted this study on a real-scene, Furnished Room, from the RAF dataset. Again, results varied by scene, with no consistent pattern (see table below). Thus, we find no correlation between a microphone’s proximity to furnitures and prediction accuracy. We added a section in the appendix about this study.
>
> | **Scene**          | **Area**                 | **T60**              | **C50**              | **EDT**                |
> |--------------------|--------------------------|----------------------|----------------------|------------------------|
> | **Apartment 1**    | Furnished  | **2.689** | 0.598 | 0.0165  |
> |     |  Spacious  |  2.810 | **0.521** |  **0.0116** |
> | **Apartment 2**    | Furnished   | **2.382**  | **0.493**  | 0.0138  |
> |    | Spacious  | 3.470 | 0.566 | **0.0102** |
> | **Furnished Room** | Furnished  | 7.630  | **0.624**  | 0.0171 |
> |  |  Spacious | **6.721** |  0.626 |  **0.0156** |
>
>
> **References:**
> [1] “AV-NeRF: Learning Neural Fields for Real-World Audio-Visual Scene Synthesis”, Liang et al., NeurIPS 2023

---

### Meta-Review · Area_Chair_KFWD · 2024-12-17

**Metareview:**

This paper proposes a method (called NeRAF) jointly learning spatial acoustic and radiance fields. NeRAF synthesizes both novel optical views and spatial audio (room impulse responses, RIR) at new positions by conditioning the acoustic field on 3D scene geometric and appearance priors from the radiance field.

This work received originally slightly contrasting reviews (5, 5, 5, 6) , and after rebuttal and discussion they reached a contrasting balance (5, 5, 6, 6). The paper was discussed by reviewers and authors and between reviewers (and AC).

Main strong points regard enhanced realism, versatility (no need to audio-visual alignment), state-of-the-art performance, and convincing experimental analysis, including ablations.

The raised weak aspects were not a few, and regarded issues in the problem setting correctness and motivations to be strengthened, especially showing how acoustic field benefits from the visual radiance field and vice versa, better justifying the claimed physical meaning of such integration. Also novelty has been questioned by a reviewer, as well as the request of additional results and comparisons. Further remarks concern the experimental analysis.

Most of these comments were addressed by the authors in an extensive way, but a couple of reviewers were still not fully convinced, even if one is possibilistic for acceptance despite the rating “marginally below threshold”. The AC read all the material and deems that authors satisfactorily replied to the most relevant comments in a reasonable way, e.g., AC thinks that novelty is not an issue in this work, there are sufficient differences wrt AV-NeRF.

In conclusion, the paper is accepted for publication to ICLR 2025.

**Additional Comments On Reviewer Discussion:**

See above

---

### Decision · Program_Chairs · 2025-01-22

Accept (Poster)